# RAST: Reasoning Activation in LLMs via Small-model Transfer

**Siru Ouyang[1], Xinyu Zhu[2], Zilin Xiao[3], Minhao Jiang[4], Yu Meng[2], Jiawei Han[1]**
[1] University of Illinois Urbana-Champaign, [2] University of Virginia
[3] Rice University, [4] GE HealthCare
siruo2@illinois.edu

## Abstract

Reinforcement learning (RL) has become a powerful approach for improving the reasoning capabilities of large language models (LLMs), as evidenced by recent successes such as OpenAI's o1 and Deepseek-R1. However, applying RL at scale remains intimidatingly resource-intensive, requiring multiple model copies and extensive GPU workloads. On the other hand, while being powerful, recent studies suggest that RL does not fundamentally endow models with new knowledge; rather, it primarily reshapes the model's output distribution to activate reasoning capabilities latent in the base model. Building on this insight, we hypothesize that the changes in output probabilities induced by RL are largely model-size invariant, opening the door to a more efficient paradigm: training a small model with RL and transferring its induced probability shifts to larger base models. To verify our hypothesis, we conduct a token-level analysis of decoding trajectories and find high alignment in RL-induced output distributions across model scales, validating our hypothesis. Motivated by this, we propose RAST, a simple yet effective method that transfers reasoning behaviors by injecting RL-induced probability adjustments from a small RL-trained model into larger models. Experiments across multiple mathematical reasoning benchmarks show that RAST substantially and consistently enhances the reasoning capabilities of base models while requiring significantly lower GPU memory than direct RL training, sometimes even yielding better performance than the RL-trained counterparts. Our findings offer new insights into the nature of RL-driven reasoning and practical strategies for scaling its benefits without incurring its full computational cost. The project page of RAST is available at https://ozyyshr.github.io/RAST/.

## 1 Introduction

Reinforcement learning (RL) [25, 62] has emerged as a powerful and prevalent paradigm for enhancing the reasoning capabilities of large language models (LLMs) [16, 51, 70, 56]. Notably, recent successes such as OpenAI's o1 model [23] and Deepseek-R1 [14] have demonstrated substantial improvements through learning from oracle-verified feedback, employing advanced RL algorithms including Proximal Policy Optimization (PPO) [55] and Group Relative Policy Optimization (GRPO) [57]. However, RL is notoriously inefficient and resource-intensive [58] — it requires loading multiple copies of the same-sized models (e.g., policy, critic, reference, reward) with extensive training GPU memory workloads. Additionally, traditional RL algorithms (e.g., PPO) typically require multiple iterations, each involving interdependent stages such as rollout, replay, and optimization [32].

On the other hand, recently, there have been a bunch of works studying behaviors of RL-trained models, aiming to answer the question, *"how does RL elicit reasoning capabilities in LLMs?"* [44, 57,

39th Conference on Neural Information Processing Systems (NeurIPS 2025).

| Model Scale | PCR |
|---|---|
| 7B | 96.03 |
| 14B | 95.60 |
| 32B | 95.22 |

**(a) PCR for different model scales**

**Math Problem:** For $0 \leq x \leq 40$ and $0 \leq y \leq 50$, find the minimum value of the following

PCR = 96.71%  expression: $\sqrt{x^2+400}+\sqrt{y^2+900}+\sqrt{x^2+y^2-80x-100y+4100}$.

**Decoding path from** $\mathcal{M}_{RL}$

… be the distance to the point (0,20), not (0,0). If `instead` `this` *(i) branch out* of we consider the expression …

`see` *(ii) self-verification*

Let's `check` the expression: $x^2+y^2-80x\ldots=(x-20)^2\ldots$ This is `incorrect` `true` *(iii) backtracking* so we need …

**(b) A case study**

■ $\mathcal{M}_{RL}$ decoding  ■ $\mathcal{M}_{base}$ decoding

Figure 1: *(a) PCR (path coverage rate) across different model scales. (b) A case study* revealing the decoding path of $\mathcal{M}_{base}$ and its RL-trained version $\mathcal{M}_{RL}$. Only a very small subset of tokens differ on the decoding path between $\mathcal{M}_{base}$ and $\mathcal{M}_{RL}$, which indicates particular reasoning behaviors.

4]. It is increasingly believed that RL does not endow LLMs with fundamentally new knowledge [74, 12]. Instead, it serves to elicit and amplify reasoning behaviors already present within base models [78, 41, 8]. For example, branching out for alternative solution paths, backtracking from incorrect steps [11, 52], and self-verification for generation [63]. The aforementioned studies lead to a major hypothesis [34] in our paper as follows:

> **Hypothesis:** RL activates latent reasoning capabilities in LLMs not by globally altering the entire output distribution, but by selectively adjusting the probabilities of a small subset of tokens that correspond to key reasoning behaviors. The majority of token probabilities — which encode core knowledge and reasoning content — remain largely unchanged.

Specifically, if RL primarily teaches models reasoning behaviors (*how* to reason) by modulating output probabilities rather than imparting fundamentally new knowledge or concepts (*what* to reason), then the adjustments learned through RL should inherently reflect reasoning skills already latent within base models. This reasoning-centric view suggests that these learned adjustments may not strongly depend on specific model scales or capacities. Consequently, this hypothesis presents a significant opportunity: applying RL to smaller, computationally efficient models and subsequently transferring the learned probabilistic adjustments to larger, more capable models. Such an approach could substantially mitigate the prohibitive computational and financial costs associated with directly performing RL on large-scale models.

We test our hypothesis via a preliminary study that compares the token-level decoding path shifts between base $\mathcal{M}_{base}$ and RL-trained models $\mathcal{M}_{RL}$ [1] of varying sizes. As illustrated in Figure 1(b), $\mathcal{M}_{RL}$ introduces only minimal shifts in the decoding path, with around 96.71% of tokens remaining unchanged in this specific case. Notably, these shifts are highly localized to a few reasoning-critical tokens, such as those triggering self-verification, branching out, or backtracking. This suggests that RL acts by amplifying latent reasoning behaviors rather than rewriting entire outputs.

Inspired by the above findings, we propose a simple and intuitive method, RAST, which leverages shifts in the output space to transfer learned "reasoning patterns" from a small RL-trained model (relative to its base) to larger base models across different scales. Extensive experimental results show that RAST substantially and consistently enhances the reasoning capabilities of base models, while requiring significantly lower GPU memory than direct RL training. Surprisingly, sometimes RAST yield even better performance than the RL-trained counterparts. Additionally, RAST also increases search space diversity compared to conventional RL training, exemplified by the superior pass@k performance. We further conduct detailed analyses on why RAST works and provide insights and practical guidelines for applying RAST, hoping to shed light on future works in this research line.

## 2  Methodology

Our goal is to enable a large base model $\mathcal{M}_{base}$ to emulate the reasoning behavior of a smaller reasoner $\mathcal{S}_{RL}$ tuned with RL, without requiring expensive RL training at scale. To this end, we begin with a preliminary study that motivates our core hypothesis and then introduce *reasoning activation via small-model transfer*, termed as RAST, a simple yet effective decoding-time method that activates reasoning capabilities across model scales.

---

[1]Unless otherwise specified, all mentions of RL-trained models in this paper refer to "RL from scratch" [76], directly training models using RL from the base model without SFT warmup.

## 2.1 Preliminary Study

We begin with a preliminary study to empirically support our *hypothesis* from Sec. 1 — RL activates reasoning capabilities in LLMs by adjusting the probabilities of *a small set of key tokens* that relate to particular reasoning behaviors. Concretely, we take the decoding path $T = [t_1, t_2, ..., t_n]$ from $\mathcal{M}_{\text{RL}}$, and feed it into $\mathcal{M}_{\text{base}}$ token by token to see if the next token prediction $t'_{i+1}$ aligns with $t_{i+1}$ (i.e., $\mathcal{M}_{\text{base}}$ generates the $t_{n+1}$-th token based on the first $n$ tokens generated by $\mathcal{M}_{\text{RL}}$, regardless of the previous $\mathcal{M}_{\text{base}}$ generated tokens):

$$t'_{i+1} = \arg\max_t P_{\mathcal{M}_{\text{base}}}(t \mid t_1, \ldots, t_i) \overset{?}{=} t_{i+1} \tag{1}$$

This setup enables us to quantify how likely $\mathcal{M}_{\text{base}}$ is to recover the RL path. This study is conducted under greedy decoding to avoid potential randomness. To capture this alignment quantitatively, we define *Path Coverage Rate (PCR)* as the proportion of tokens in $T$ for which the base model exactly matches the RL output:

$$\text{PCR}(T) = \frac{1}{n-1} \sum_{i=1}^{n-1} \mathbb{I} \left[ t'_{i+1} = t_{i+1} \right] \tag{2}$$

A high PCR indicates that the $\mathcal{M}_{\text{base}}$ is already well-aligned with the RL decoding path, with only minor adjustments needed to activate desired reasoning behaviors. In our implementation, Qwen-2.5-32B-SimpleRL-Zoo serves as $\mathcal{M}_{\text{RL}}$ and Qwen2.5-32B is used as $\mathcal{M}_{\text{base}}$. We randomly sampled 50 trajectories from $\mathcal{M}_{\text{RL}}$ from MATH500 [18] and took the sample-level average as the final results. As shown in Figure 1(a), we observe that PCR remains remarkably high ($> 95\%$) across all model scales, indicating that RL-induced distributional shifts are notable only on a small set of tokens, with the majority of tokens also predictable by $\mathcal{M}_{\text{base}}$. Additionally, we found that the disparities mainly come from tokens that reflect certain reasoning behaviors. This indicates that by steering around these key tokens, $\mathcal{M}_{\text{base}}$ is able to recover the reasoning path generated by the RL-trained model.

## 2.2 RAST: Reasoning Activation in LLMs via Small-model Transfer

Building on the findings from our preliminary study, we hypothesize that minor, targeted adjustments to the $\mathcal{M}_{\text{base}}$'s output distribution can effectively enable it to perform on par with its RL-trained counterpart, without the need for expensive RL optimization directly conducted upon the large base model. In other words, we aim to transform $\mathcal{M}_{\text{base}}$ into a stronger reasoner at inference time by activating the latent reasoning capabilities already present within it in the token/output space.

To this end, we propose RAST, a decoding-time method that activates reasoning capabilities in large models by transferring logit-level adjustments from smaller RL-tuned models. Given the smaller model pair $\mathcal{S}_{\text{base}}$ and $\mathcal{S}_{\text{RL}}$, we propose leveraging their differences in logit distributions as reusable reasoning correction signals. Specifically, at the decoding time stamp $t$, with previous input as $x_{<t}$, we compute the logit scores of $\mathcal{M}_{\text{base}}, \mathcal{S}_{\text{base}}, \mathcal{S}_{\text{RL}}$, and define the final probability distribution over tokens for the enhanced model $\tilde{\mathcal{M}}$ as:

$$P_{\tilde{\mathcal{M}}}(X_t \mid x_{<t}) = \text{softmax} \left[ \mathcal{M}_{\text{base}}(X_t \mid x_{<t}) + \lambda(\mathcal{S}_{\text{RL}}(X_t \mid x_{<t}) - \mathcal{S}_{\text{base}}(X_t \mid x_{<t})) \right] \tag{3}$$

where the adjustment terms represent the difference in token-level scoring between the RL-trained model and its base counterpart, denoted as $\Delta R$. $\lambda$ controls the strengths of $\Delta R$. These logits encode reasoning-oriented shifts that can be transferred to the larger base model, allowing it to mimic improved inference behavior without retraining. Figure 2 outlines the overview of RAST, showing how $\Delta R$ from a small RL-tuned model is injected to adjust the output distribution of $\mathcal{M}_{\text{base}}$ at decoding time, selectively amplifying reasoning-relevant tokens (e.g., "instead") while preserving base predictions (e.g., "of") elsewhere. The additive formulation enables lightweight adaptation at inference time by altering the output distribution in a way that reflects reasoning preferences learned by the smaller model.

Our method is conceptually related to [36, 37, 31], which apply similar manipulation in logits spaces for generation steering. However, while these methods are often task-specific or used for controlling style/toxicity, our focus is on eliciting complex reasoning behavior, and we demonstrate that even small-scale reasoning signals can be reliably transferred across model sizes and domains.

## 3 Unlocking Reasoning Activation

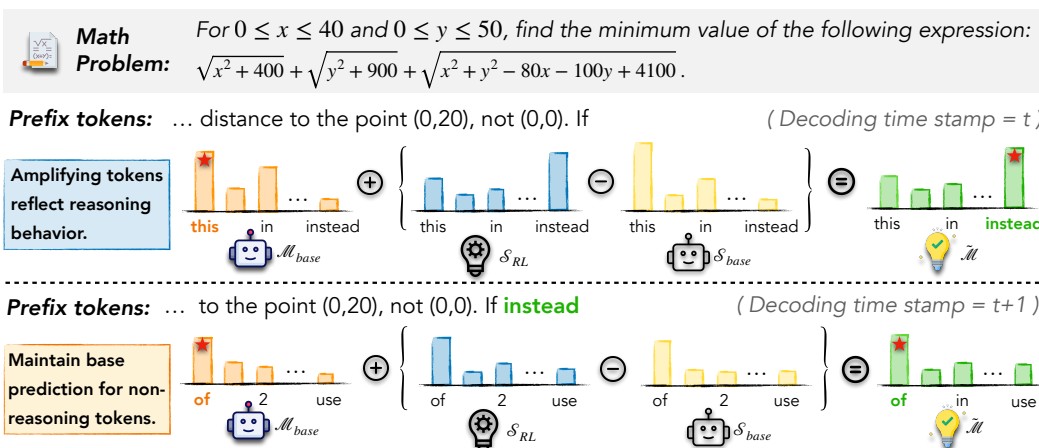

Figure 2: A concrete illustration of RAST: logit differences from a small RL-tuned model $\mathcal{S}_{\text{RL}}$ guide a large base model $\mathcal{M}_{\text{base}}$ at decoding time, amplifying reasoning-relevant predictions (e.g., "instead") while maintaining base outputs for non-reasoning tokens (e.g., "of").

## 3.1 Experimental Setup

**Models and Tasks** We systematically evaluate model performance across a comprehensive suite of mathematical reasoning tasks of varying difficulty, including standard benchmarks such as MATH500 [18], Minerva [30], OlympiadBench [17], GSM8K [6], as well as competition-level benchmarks AIME24 and AMC23. Our primary models are from the Qwen-2.5 family (1.5B, 7B, 14B, and 32B) [71] alongside their corresponding RL-trained variants using SimpleRL-Zoo [76] (e.g., SimpleRL-7B denotes the "RL from scratch" variant of Qwen-2.5-7B). We further demonstrate the generalizability of RAST across different model architectures and downstream tasks. To validate this, we conduct experiments using the Llama-3.1 series (8B, 70B) [13] and their zero RL-trained counterparts. Additionally, we assess model performance on coding tasks, utilizing zero RL-trained models from Code-R1 [40] trained from Qwen-2.5-1M [72] on coding benchmarks including HumanEval [2], MBPP+ [1], and LiveCodeBench [24]. For details of datasets and setup used in our experiments, please refer to Appendix A. We also present additional experiments on MMLU [18] and GPQA [54] and analysis results in Appendix C.

**Evaluation Metrics** Following previous work [19], and to ensure rigorous evaluation across models and tasks, we perform inference runs $k$ up to 32 for each experimental setup and report the following metrics calculated over the collected trajectories:

- **Pass@$k$**: Pass@$k$ evaluates whether at least one correct solution is found among $k$ sampled outputs per problem. Formally,

$$\text{Pass@}k = \frac{1}{N} \sum_{i=1}^{N} \mathbb{I} \left( \sum_{j=1}^{k} \mathbb{I}\left(\hat{y}_{i,j} = y_i\right) \geq 1 \right), \tag{4}$$

where $N$ is the total number of problems, $\hat{y}_{i,j}$ is the prediction for the $i$-th problem in the $j$-th run, $y_i$ is the corresponding ground truth, and $\mathbb{I}(\cdot)$ denotes the indicator function.[2]

- **Recovery Rate**: The recovery rate quantifies how much of the gap between the base model and a stronger RL-tuned model is recovered by the proposed method. Formally,

$$\text{Recovery Rate} = \frac{\text{Accuracy}_{\text{RAST}} - \text{Accuracy}_{\text{Base}}}{\text{Accuracy}_{\text{RL}} - \text{Accuracy}_{\text{Base}}}, \tag{5}$$

where "Accuracy" denotes the averaged pass@1 over 32 runs. Higher values of *recovery rate* indicate a more effective recovery of the performance gap.

**Decoding Configurations** To accelerate the inference speed, we implemented a revised vLLM [27] version to support RAST. For mathematical reasoning, decoding is performed using a temperature

---

[2]We use the same answer extraction and matching for performance evaluation as SimpleRL, following https://github.com/hkust-nlp/simplerl-reason/tree/v1/examples/simplelr_math_eval.

Table 1: Experiment results of RAST on mathematical reasoning datasets with Qwen-2.5 model series. All numbers are computed across 32 runs with sampling, except that all base models use greedy decoding. *Avg.* indicates the averaged *pass@1* over 32 runs and *RR.* denotes the recovery rate.

| Models | Math500 | | AIME24 | | AMC | | Minerva | | Olympiad | | GSM8K | |
|---|---|---|---|---|---|---|---|---|---|---|---|---|
| | *Avg.* | *RR.* | *Avg.* | *RR.* | *Avg.* | *RR.* | *Avg.* | *RR.* | *Avg.* | *RR.* | *Avg.* | *RR.* |
| Qwen-2.5-32B | 68.6 | - | 3.3 | - | 52.5 | - | 21.0 | - | 33.1 | - | 93.1 | - |
| $+\Delta R_{1.5B}$ [†] | 73.7 | 40.2% | 12.5 | 37.9% | 54.7 | 12.9% | 25.1 | 84.1% | 36.7 | 30.5% | 93.3 | 7.7% |
| $+\Delta R_{7B}$ | 79.0 | 81.9% | 16.4 | 53.9% | 58.4 | 34.5% | 32.0 | 87.3% | 41.7 | 72.9% | 94.4 | 50.0% |
| $+\Delta R_{14B}$ | 80.7 | 95.3% | 18.3 | 61.7% | 65.2 | 74.3% | 34.2 | 104.8% | 43.5 | 88.1% | 95.3 | 84.6% |
| SimpleRL-32B | 81.3 | - | 27.6 | - | 69.6 | - | 33.6 | - | 44.9 | - | 95.7 | - |
| Qwen-2.5-14B | 63.8 | - | 6.7 | - | 47.5 | - | 19.1 | - | 31.1 | - | 91.4 | - |
| $+\Delta R_{1.5B}$ [†] | 70.3 | 45.1% | 10.9 | 54.5% | 50.3 | 25.0% | 22.9 | 29.0% | 35.5 | 37.0% | 92.4 | 31.3% |
| $+\Delta R_{7B}$ | 77.4 | 94.4% | 16.2 | 123.4% | 56.1 | 76.8% | 28.8 | 74.0% | 40.0 | 75.4% | 93.3 | 59.4% |
| SimpleRL-14B | 78.2 | - | 14.4 | - | 58.7 | - | 32.2 | - | 42.9 | - | 94.6 | - |
| Qwen-2.5-7B | 62.9 | - | 6.7 | - | 32.5 | - | 19.9 | - | 28.0 | - | 87.7 | - |
| $+\Delta R_{1.5B}$ [†] | 68.4 | 41.0% | 9.5 | 31.5% | 41.5 | 38.5% | 23.0 | 51.7% | 34.6 | 57.4% | 89.8 | 50.0% |
| SimpleRL-7B | 76.3 | - | 15.6 | - | 55.9 | - | 25.9 | - | 39.5 | - | 91.9 | - |

[†] indicates that the prompt used for RAST does not match the one used to train the small RL-tuned model (see Figure 7), due to the training inconsistency for available RL-tuned models (e.g., from SimpleRL-Zoo). As a result, the transferred $\Delta R_{1.5B}$ yields relatively smaller gains.

Table 2: Experiment results of RAST on mathematical reasoning datasets. $\Delta R$ is borrowed from Llama-3.1-8B-SimpleRL-Zoo [76]. Numbers are computed on 32 runs with sampling.

| Models | Math500 | AIME24 | AMC | Minerva | Olympiad | GSM8K |
|---|---|---|---|---|---|---|
| Llama-3.1-70B | 26.2 | 0.0 | 12.5 | 10.3 | 5.9 | 56.1 |
| $+\Delta R_{8B}$ | 33.1 | 2.8 | 16.9 | 12.6 | 7.1 | 82.2 |

Table 3: Experiment results of RAST on code reasoning tasks with the Qwen-2.5-14B-1M model as $\mathcal{M}_{base}$ and $\Delta R$ from Code-R1-Zero [40] using greedy decoding.

| Models | HumanEval+ | MBPP+ | LiveCodeBench | Average |
|---|---|---|---|---|
| Qwen-2.5-14B-1M | 82.3 | 69.6 | 34.3 | 62.1 |
| $+\Delta R_{7B}$ | 85.4 | 76.5 | 37.6 | 66.5 |

setting of 1.0 and nucleus sampling with a top-$p$ of 0.95, allowing a maximum generation length of $16,384$ tokens, consistent with prior work [76]. We set $\lambda$ in Equation 3 to 1.0 for all experiments. For code reasoning tasks, we follow previous evaluation settings [40, 24] and use greedy decoding. Specifically, we use EvalPlus [38, 39] for HumanEval+ and MBPP+. Our experiments are conducted over 8 NVIDIA A6000 GPUs on a single node, with GPU utilization and tensor parallelism parameters dynamically adjusted based on the model size. Typically, inference requires approximately 30 minutes per run for larger datasets like MATH500 and GSM8K, whereas smaller datasets such as AIME24 and AMC23 complete within 3–5 minutes per run. For specific parameters used for each benchmark, please refer to Appendix B.

## 3.2 Main Results

**RAST enables consistent and scalable reasoning gains.** Table 1 summarizes the performance of RAST on six mathematical reasoning benchmarks using the Qwen-2.5 model family at 1.5B, 7B, 14B, and 32B scales. Across all settings, RAST delivers substantial improvements over the base models in both averaged pass@1 and the corresponding recovery rate. Notably, with signals from smaller RL-trained models, RAST can even approach or beat the performance of RL-trained counterparts of the base model. For instance, applying $\Delta R_{14B}$ to the 32B base model achieves approximate or superior results on MATH500, Minerva, and GSM8K compared with the 32B RL-trained model. These findings validate the effectiveness of RAST in enhancing reasoning capabilities at inference time, without any retraining or RL on the target model.

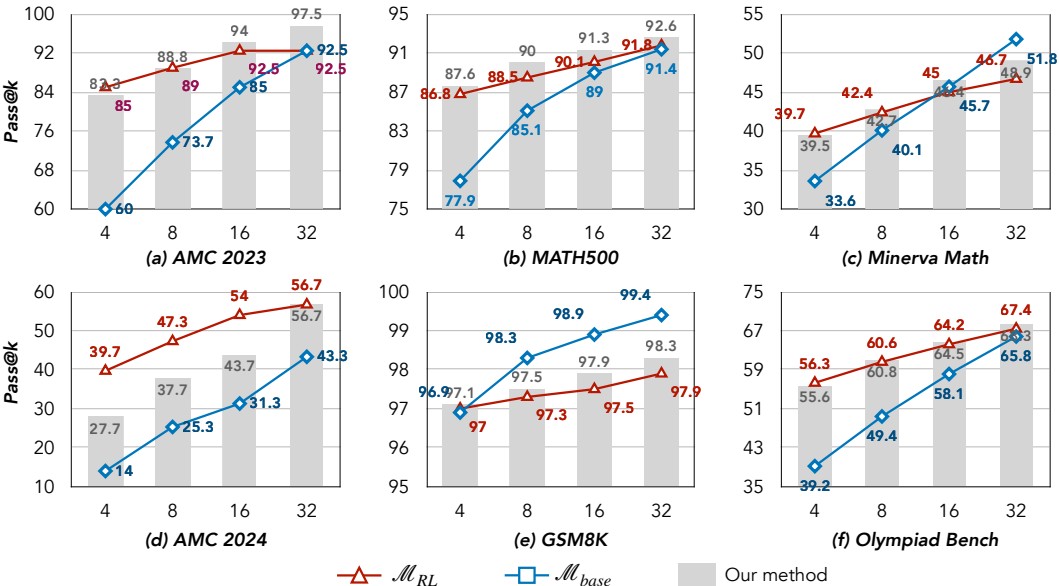

Figure 3: The illustration of pass@$k$ for different values of $k$ on 6 mathematical reasoning datasets, where $\mathcal{M}_{\text{base}}$ is Qwen-2.5-32B, RAST uses $\Delta R_{14B}$, and $\mathcal{M}_{\text{RL}}$ is the RL-trained version of $\mathcal{M}_{\text{base}}$.

$\Delta R$ **from stronger experts yield greater gains.** The effectiveness of RAST also depends on the strength of the $\mathcal{S}_{\text{RL}}$ and $\mathcal{S}_{\text{base}}$ that generates the delta logit $\Delta R$. For each base model, using larger delta sources (e.g., base model of 32B with $\Delta R_{7B}$ or $\Delta R_{14B}$ compared with $\Delta R_{1.5B}$) leads to greater improvement. Taking the 32B base on MATH500 as an example, accuracy increases progressively from 73.7 (with $\Delta R_{1.5B}$) to 80.7 (with $\Delta R_{14B}$), while the ceiling model, or the upper bound reaches 81.3. Similarly, $\Delta R_{7B}$ also works better than $\Delta R_{1.5B}$ for the 14B base model across all datasets. This trend suggests that logit deltas encode richer reasoning signals as the $\mathcal{S}_{RL}$ model scale increases, making RAST a flexible tool for knowledge transfer across various model scales.

**Trade-off between $\mathcal{M}_{\text{base}}$ and $\Delta R$.** The effectiveness of RAST also depends on the capacity of the base model $\mathcal{M}_{\text{base}}$ and its alignment with $\Delta R$. In general, stronger base models exhibit higher recovery rates, indicating greater receptiveness to transferred reasoning signals. For example, when applying RAST to the 32B base, the recovery rate is often higher than when using 14B or 7B. On GSM8K, RAST bridges nearly the entire gap between the base model (93.1) and the RL expert (95.7), achieving 95.3 with the delta logit from a 14B model. In contrast, for the 7B base, the gain is smaller (87.7 to 91.9), even though the same $\Delta R_{7B}$ is applied. This suggests that higher-capacity base models are more receptive to the transferred reasoning signal. However, increasing base model capacity alone does not guarantee better outcomes. When applying $\Delta R_{7B}$ to both 14B and 32B bases, the 14B base yields a higher recovery rate, suggesting that a large capability gap between the base and $\Delta R$ may hinder effective transfer. These observations highlight a trade-off: while stronger base models benefit more from compatible deltas, excessively mismatched pairs may reduce the efficacy of reasoning activation.

### 3.3 RAST Boosts Reasoning Diversity

Figure 3 illustrates the pass@$k$ accuracy trends across six mathematical reasoning benchmarks, using the Qwen-2.5-32B base model augmented with $\Delta R_{14B}$. For each problem, we randomly sample $k$ outputs from a 32-sample pool, repeat this process 10 times, and report the average accuracy. This setup allows us to assess how well RAST supports diverse solution trajectories under varying sampling sizes. Based on the results, we have the following key observations:

**Increasing $k$ consistently improves accuracy.** Across all benchmarks, pass@$k$ increases monotonically with larger $k$, confirming the benefit of sampling multiple decoding paths. This trend reflects that RAST promotes solution diversity—a larger $k$ enables broader exploration of plausible answers, increasing the likelihood of capturing correct responses even when individual generations are imperfect. We also noticed that on most benchmarks except GSM8K, pass@$k$ for $\mathcal{M}_{\text{RL}}$ is large than $\mathcal{M}_{\text{base}}$, which might result from GSM8K's simplicity.

**Pass@$k$ surpasses the ceiling performance.** Remarkably, in all benchmarks, RAST achieves pass@$k$ accuracy that equals or even exceeds the performance of $\mathcal{M}_{\text{RL}}$. This stands in contrast to prior findings [57, 74], which reported limited or deteriorated performance in pass@$k$ under RL training. We posit that this effect may stem from the implicit ensembling of knowledge across models, which enhances diversity in the search space. Additionally, RAST can sometimes beat the pass@$k$ of $\mathcal{M}_{\text{base}}$ on benchmarks like AMC, MATH500, and Olympiad Bench. This surprising behavior suggests that RAST not only saves the costly training efforts of RL, but also introduces a distinct form of diversity or guidance in the sampling space that helps recover correct answers, a notable drawback of RL-trained models.

### 3.4 RAST Generalizes Well to Other Models and Tasks

We conduct experiments of RAST on another model family, Llama-3.1, and on additional code reasoning tasks. The results are shown in Tables 2 and 3. Applying RAST to Llama-3.1-70B using $\Delta R_{8B}$ from a smaller RL-tuned model [76] yields consistent gains across all six mathematical reasoning datasets. As shown in Table 2, we observe a $+2.8$ absolute improvement on AIME24, $+4.4$ on AMC, and $+2.3$ on Olympiad. We also found that the improvement in MATH500 and GSM8K was particularly high. This might be due to the training recipe of $S_{RL}$, where they are trained with problems from MATH500 and GSM8K. Nonetheless, the experiments demonstrate that RAST is effective for different model families apart from Qwen, reaffirming the hypothesis that reasoning-relevant distributional shifts are transferrable and model-agnostic. In Table 3, we evaluate RAST on Qwen-2.5-14B-1M [72] using $\Delta R$ from its 7B counterpart [40] on three code reasoning benchmarks. RAST achieves consistent improvements on all datasets, leading to an overall $+4.4$ absolute improvement in average performance. This indicates that the method is not limited to mathematical reasoning, but code-related reasoning tasks. These findings confirm the broad applicability of RAST across model architectures, parameter scales, and reasoning domains.

## 4 Understanding Reasoning Activation

### 4.1 Similarity of $\Delta R$ as Signal for Transferability

As shown in Table 1, model performance varies widely across settings, motivating the need to understand what governs effective transferability. The delta logits $\Delta R$ (defined in Equation 3) capture the modification induced by the reasoning-enhanced model relative to its base. To quantify the alignment between these delta logits across different model pairs, we adopt *cosine similarity* as the measure. This choice is motivated by prior work [61, 15], which demonstrates that cosine similarity effectively captures directional alignment in high-dimensional spaces, independent of magnitude. Following Section 2, we randomly sample 50 examples from MATH500 and extract the decoding trajectory $T$ (with tokens $t$) generated by SimpleRL-32B. Each trajectory is then fed through $\mathcal{M}_{\text{base}}$ and $\mathcal{M}_{\text{RL}}$ token by token to compute $\Delta R$ with prefix $T_{<t}$. We then log the average cosine similarity across the trajectory as:

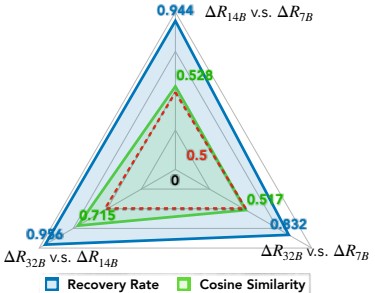

Figure 4: Cosine similarity vs. recovery rate across delta logit pairs ($\Delta R$) from varying model scales. E.g., "$\Delta R_{14B}$ v.s. $\Delta R_{7B}$" denotes AvgCosineSim($\Delta R_{14B}, \Delta R_{7B}$).

$$\text{AvgCosineSim}(\Delta R_1, \Delta R_2) = \frac{1}{T} \sum_{t=1}^{T} \frac{\Delta R_1^{(t)} \cdot \Delta R_2^{(t)}}{\|\Delta R_1^{(t)}\| \, \|\Delta R_2^{(t)}\|} \tag{6}$$

The results are shown in Figure 4. We found that the recovery rate exhibits a positive correlation with cosine similarity—as $\Delta R$ between models becomes more aligned, transferability improves.

### 4.2 Empirical Token-Level Signals of Reasoning Activation

To further examine how RAST elicits reasoning behaviors during generation, we perform a token-level analysis focusing on linguistic traces that reflect three hallmark reasoning behaviors: (i) *branching out*, (ii) *backtracking*, and (iii) *self-verification* as previously mentioned in Section 1. For each behavior,

we manually curated a set of representative tokens based on prior qualitative observations and related work [11, 52, 63]. These tokens serve as behavioral signatures that may surface when the model engages in complex reasoning. The complete set of curated tokens is displayed in Figure 9. We also present a case study of token-level reasoning activation for a more intuitive view in Appendix C.5.

We compute the frequency of these tokens in all 32 model output trajectories across six mathematical reasoning benchmarks, comparing the base model $\mathcal{M}_{base}$, RL-trained ceiling model $\mathcal{M}_{RL}$, and RAST with $\Delta R_{14b}$ applying to 32B base models. Figure 5 presents the normalized occurrence rates of each token category across different models. Two key trends emerge:

**Dataset-specific reasoning emphasis.** Different benchmarks accentuate different types of reasoning behaviors. For instance, AIME 2024 and Olympiad Bench exhibit a pronounced increase in *branching out* and *backtracking* tokens, indicating that these tasks may demand exploring alternative solution paths and revisiting previous steps. In contrast, datasets like AMC 2023 and MATH500 emphasize *self-verification*, with higher frequencies of verification-related tokens such as "check" or "confirm". This variation suggests that reasoning demands are not uniform across benchmarks, and token-level analysis can surface behavior-specific task signals.

**RAST steers base models closer to RL-trained behaviors.** Across all datasets and reasoning categories, we observe that RAST consistently increases the frequency of reasoning-related tokens relative to $\mathcal{M}_{base}$ and more closely matches the RL-trained model $\mathcal{M}_{RL}$. This alignment is particularly clear in Minerva, Olympiad Bench, and GSM8K, where RAST nearly mirrors the behavior of $\mathcal{M}_{RL}$ in the *self-verification* dimension. These results support our central claim that the delta logit signal $\Delta R$ effectively induces reasoning traits without the need for full RL training on large-scale models.

Together, these findings offer empirical evidence that RAST not only activates latent reasoning capabilities within base models but also tailors such activation in a task-sensitive manner, approximating the behavioral signature of much costlier RL-trained experts.

## 4.3 Efficiency Analysis

**GPU Memory Overhead.** We first present the efficiency analysis in terms of estimated GPU memory requirements, highlighting the computational advantage of RAST over conventional RL training. We summarize the results in Table 4. We estimate the memory overhead in terms of GPU memory used for RAST (including training $\mathcal{S}_{RL}$ and inference cost), and $\mathcal{M}_{RL}$. The results are estimated considering tensor parallel and CPU offloading (since these are common tricks during training) based on the following dimensions: (i) model memory footprint (FP16), (ii) optimizer states,

Table 4: Comparison of memory overhead and inference speed between our approach RAST and the conventional RL training pipeline (i.e., GRPO). We also report the averaged performance recovery rate (RR.) for all mathematical reasoning benchmarks.

| Settings | 32B | | 32B | | 14B | |
| --- | --- | --- | --- | --- | --- | --- |
| | $\mathcal{M}_{base}$ | $\mathcal{M}_{RL}$ | $\mathcal{M}_{base}$ | $\mathcal{M}_{RL}$ | $\mathcal{M}_{base}$ | $\mathcal{M}_{RL}$ |
| | $+\Delta R_{14B}$ | - | $+\Delta R_{7B}$ | - | $+\Delta R_{7B}$ | - |
| *Memory Overhead* | | | | | | |
| GPU memory | ~160G | ~350G | ~100G | ~350G | ~100G | ~160G |
| # GPU cards | 4×80G | 8×80G | 2×80G | 8×80G | 2×80G | 4×80G |
| *Inference Speed* | | | | | | |
| Input Tokens | 61.1 | 73.0 | 62.4 | 73.0 | 65.0 | 75.1 |
| Output Tokens | 200.3 | 243.1 | 202.7 | 243.1 | 216.1 | 249.5 |
| Averaged RR. | 84.8% | 100% | 63.4% | 100% | 83.9% | 100% |

and (iii) activations & buffers. Details of the computation for estimation could be found in Appendix D. Despite using significantly fewer resources, RAST achieves high recovery rates across all settings (e.g., reaching over 84% in the most demanding 32B + $\Delta R_{14B}$ configuration). This demonstrates that RAST retains most of the performance benefits of full-scale RL training while reducing the computational burden by up to 50% in terms of GPU memory and hardware requirements.

**Inference Speed.** Table 4 further offers the actual inference speed in terms of the number of input and output tokens processed per second for RAST's practicality. We observe that RAST incurs around 13-16% latency at inference time compared to directly using $\mathcal{M}_{RL}$ for decoding at different scales. This is expected and acceptable due to the need to compute multiple model logits at each decoding step. We believe that this latency could be further optimized by efficient decoding methods such as speculative decoding [29, 69, 49].

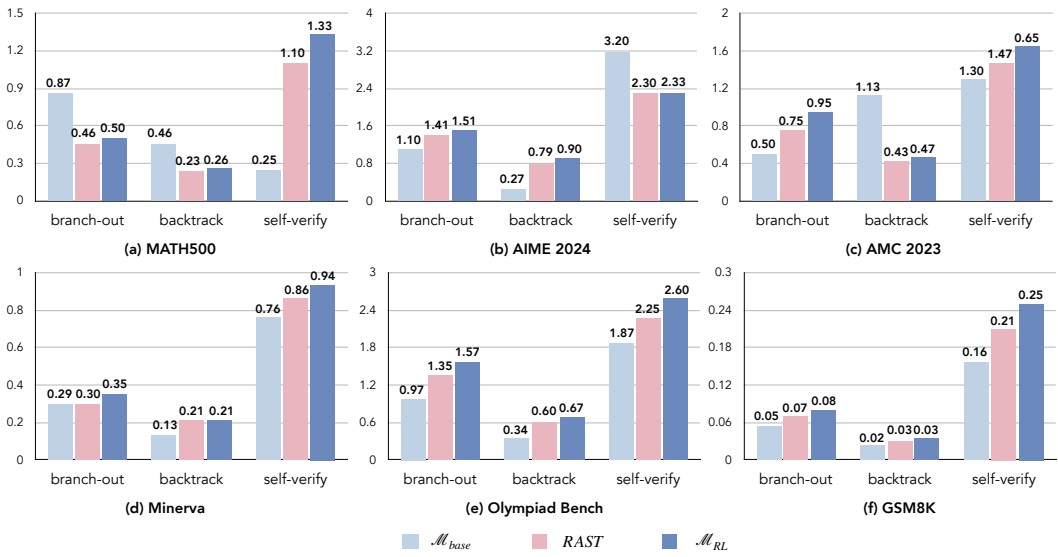

Figure 5: Normalized frequencies of reasoning-related tokens across three models ($\mathcal{M}_{\text{base}}$, RAST, and $\mathcal{M}_{\text{RL}}$) over six benchmarks. Each subfigure reflects a different reasoning behavior category.

## 4.4 Robustness regarding $\tau$ and $\lambda$

To evaluate the sensitivity of our method to decoding-time hyperparameters, we perform a grid search over sampling temperature $\tau$ and $\lambda$ used in Equation 3. The results are summarized with performance trends visualized in Figure 6. We plot accuracy across varying $\tau$ values (with $\lambda = 0.5$ fixed) and varying $\lambda$ values (with $\tau = 1.0$ fixed), including standard deviation error bars to reflect stability across multiple runs. Note that we use these fixed values for investigation since they represent the peak performance. In both settings, the accuracy remains consistently high within a reasonable range. Specifically, with $\tau \in [0.5, 1.0]$ and $\lambda \in [0.3, 1.5]$, the performance would be reasonably good within a certain range. These trends demonstrate that our method is robust to moderate fluctuations in decoding-time hyperparameters and does not rely on precise tuning for strong performance.

## 5 Related Work

### 5.1 Reinforcement Learning for LLM Complex Reasoning

Reinforcement learning (RL) has emerged as an important process of LLM post-training for human preference alignment. Early successes such as reinforcement learning with human feedback (RLHF) [48] demonstrated the effectiveness of reward modeling from preference data [75, 7]. Building upon this, follow-up works have explored RL as a means to directly optimize for reasoning quality [23] with outcome supervision [28] or process supervision [33, 64] as verifiable rewards, specifically in the domain of math [57] and coding [66, 40].

More recently, the success of DeepSeek-R1 [14] introduced a notable shift in training methodology with the "zero-RL" paradigm, where RL is applied directly to the base LLM, entirely bypassing intermediate supervised fine-tuning. Following its release, the open-source community has made significant strides in replicating [76, 50, 81], extending [73], and interpreting [78, 74, 42] the R1 algorithm and its behavioral consequences. Our work builds directly upon these open-source Zero-RL models and takes a step further by exploring whether the reasoning behaviors elicited through RL in small models can be transferred to larger base models without additional RL training. Specifically, we focus on leveraging the output distribution (logits) of small RL-trained reasoning models to activate similar behaviors in larger models across scales.

### 5.2 Decoding-time Strategy in Language Models

Decoding-time methods [59] have been explored largely in text generation, typically by manipulating the logit distribution from base language models. Earlier research efforts focused on sampling-based

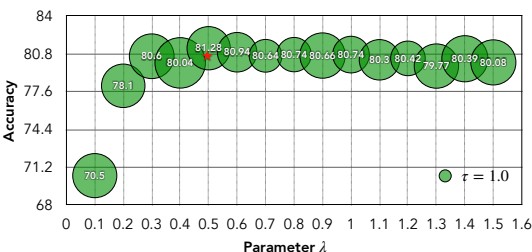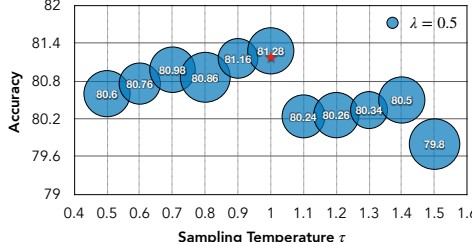

Figure 6: Experiment results with varying $\lambda$ (left) and $\tau$ (right) on MATH500 dataset, where ★ denotes the peak performance. The position represents the accuracy, and the size of the circle denotes the standard deviation over 32 runs.

strategies aimed at improving generation quality through techniques like hierarchical decoding [10], nucleus sampling [20], and locally typical sampling [45]. More recent approaches incorporate the notion of *contrastiveness*. Specifically, Contrastive Decoding [31] exploits the disagreement between a stronger (expert) and a weaker (amateur) model to down-weight completions favored by the weaker model. DExperts [37] steers generation by contrasting models fine-tuned on desirable and undesirable attributes. Proxy-tuning [36] and emulated tuning [46] further leverage the idea for training-free alignment, while DoLA [5] contrasts different layers in next-token-prediction for factuality. The philosophy of contrastiveness has since been extended to a variety of downstream tasks, including machine translation [65], retrieval-augmented generation (RAG) [53], and conflict resolution in knowledge-intensive tasks [60]. While these works primarily focus on stylistic control or factual consistency in open-ended text generation, our approach targets reasoning-specific improvements.

When it comes to reasoning, [47] explores contrastive decoding for mathematical reasoning, while [35] identifies critical tokens in the reasoning trajectory using contrastive estimation. Our work builds on this growing line of decoding-time reasoning enhancement. Specifically, we leverage the divergence between Zero-RL-trained experts and their base counterparts to guide decoding, aiming to surface and amplify reasoning behaviors. In contrast to prior approaches that operate on instruction-tuned or supervised models, our method explicitly targets RL-induced reasoning signals and explores their transferability across model scales.

## 6    Conclusion and Discussion

We introduce RAST, a decoding-time framework that enables scalable reasoning enhancement in LLMs by transferring logit-level guidance from smaller RL-tuned models. Comprehensive experiments show that RAST consistently boosts the performance, often approaching or surpassing the performance of much more expensive ceiling models. Further analysis confirms the robustness of our method across decoding hyperparameters and highlights the alignment between delta signals and reasoning activation. Our findings suggest a practical and efficient pathway for eliciting complex reasoning in LLMs, opening new avenues for decoding-time reasoning enhancement and model behavior study for RL. We also present some insights toward future directions in Appendix E.

## Acknowledgments and Disclosure of Funding

We sincerely thank the anonymous reviewers and the area chair for their constructive feedback and valuable suggestions, which greatly helped improve this work. Research was supported in part by National Science Foundation IIS-19-56151, NSF IIS 25-37827, the Molecule Maker Lab Institute: An AI Research Institutes program supported by NSF under Award No. 2019897, and the Institute for Geospatial Understanding through an Integrative Discovery Environment (I-GUIDE) by NSF under BRIES Program No. HR0011-24-3-0325. The research has used the Delta/DeltaAI advanced computing and data resource, supported in part by the University of Illinois Urbana-Champaign and through allocation #250851 from the Advanced Cyberinfrastructure Coordination Ecosystem: Services & Support (ACCESS) program, which is supported by National Science Foundation grants OAC 2320345, #2138259, #2138286, #2138307, #2137603, and #2138296. Any opinions, findings, and conclusions or recommendations expressed herein are those of the authors and do not necessarily represent the views, either expressed or implied, of DARPA or the U.S. Government.

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

# A  Implementation Details

## A.1  Datasets

We provide the details for all the datasets used in our work as follows. All datasets or benchmarks used in this paper are publicly available online. For mathematical reasoning tasks, we include 6 widely used datasets, detailed below:

**MATH500**   The original MATH collection contains 12,500 problems in total, with 8,000 training and 4,500 test problems, meticulously curated to cover a wide range of topics and difficulty levels. Each problem in MATH has a full step-by-step solution that can be used to teach models to generate answer derivations and explanations. MATH500 (could be found at https://huggingface.co/datasets/HuggingFaceH4/MATH-500) is a non-standard train/test split of the original MATH dataset [18], following [33] to avoid the risk of over-fitting and for more efficient testing configurations. These 500 test problems are selected uniformly at random, and are representative of the test set as a whole.

**GSM8K**   GSM8K (Grade School Math 8K) [6] is a dataset of 8,500 high-quality linguistically diverse grade school math word problems. The dataset was created to support the task of question answering on basic mathematical problems that require multi-step reasoning. The test set of GSM8K (could be found at https://huggingface.co/datasets/openai/gsm8k) includes 1,319 problems in total.

**Olympiad Bench**   Olympiad Bench [17] is originally an Olympiad-level bilingual multimodal scientific benchmark, which contains 8,952 math and physics questions from international Olympiads, Chinese Olympiads, Chinese college entrance examinations, and mock exams. To support our testing needs, we select a subset from the Olympiad Bench that is categorized as "open-ended", "text-only" and "competition-level". Together, there are 675 test problems (could be found at https://huggingface.co/datasets/Hothan/OlympiadBench/viewer/OE_TO_maths_en_COMP) for this subset used in our paper.

**AIME**   The collection of AIME actually contains problems from the American Invitational Mathematics Examination (AIME). AIME is a prestigious high school mathematics competition known for its challenging mathematical problems. In our work, we follow previous works and adopt all the 30 problems (could be found at https://huggingface.co/datasets/HuggingFaceH4/aime_2024) in AIME 2024 for testing purposes.

**AMC**   Similar to AIME, AMC is another very challenging dataset that contain problems in competitions, specifically, the American Mathematics Competitions (AMC). The collection of AMC actually contains 40 problems (could be found at https://huggingface.co/datasets/math-ai/amc23) in total in the year of 2023 for our testing set.

**Minerva**   The testing set of Minerva math is curated in [30], which consists of STEM problems at the undergraduate level. In total, there are 272 problems (could be found at https://huggingface.co/datasets/math-ai/minervamath), 191 of which have numeric solutions and 81 have symbolic solutions.

For code reasoning tasks, we incorporate three datasets as following:

**HumanEval+**   HumanEval+ is an adapted version from the original HumanEval by [38]. HumanEval+ extends beyond HumanEval with additional high-quality and automatically generated test inputs to $80\times$, powered by both LLM- and mutation-based strategies. It contains 164 samples for testing (could be found at https://huggingface.co/datasets/evalplus/humanevalplus).

**MBPP+**   HumanEval+ is also an adapted version from the original MBPP by [38]. The construction process is quite similar to HumanEval+, and it contains 378 samples for the testset (could be found at https://huggingface.co/datasets/evalplus/mbppplus).

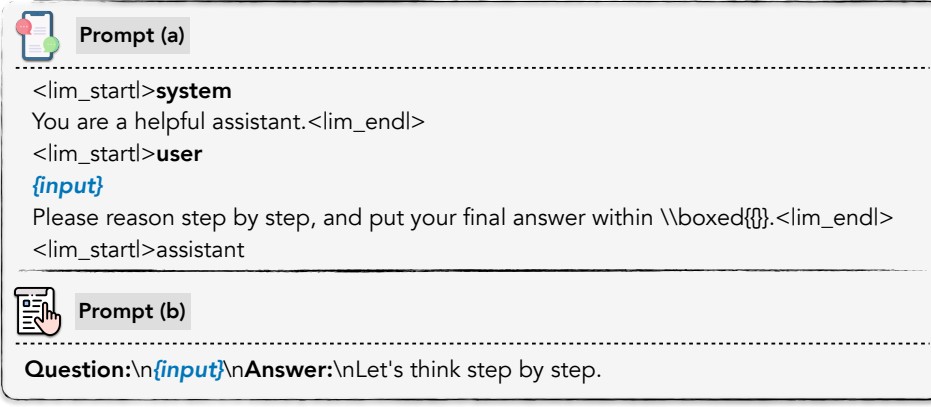

Figure 7: Prompt templates used for mathematical reasoning tasks.

Table 5: Detailed decoding configurations (hyperparameters) used in our experiments.

| Settings | gpu_utilization | tensor_parallel_size | temperature | $\lambda$ | top_p | max_seq_len |
|---|---|---|---|---|---|---|
| Qwen-2.5-32B | 0.6 | 4 | 0.0 | | 1.0 | |
| $+\Delta R_{1.5B}$ | 0.6 | 4 | | | | |
| $+\Delta R_{7B}$ | 0.75 | 4 | | 1.0 | 0.95 | 16,384 |
| $+\Delta R_{14B}$ | 0.75 | 4 | 1.0 | | | |
| 32B-RLZero | 0.6 | 4 | | | | |
| Qwen-2.5-14B | 0.6 | 2 | 0.0 | | 1.0 | |
| $+\Delta R_{1.5B}$ | 0.6 | 2 | | 1.0 | 0.95 | 16,384 |
| $+\Delta R_{7B}$ | 0.75 | 2 | 1.0 | | | |
| 14B-RLZero | 0.6 | 2 | | | | |
| Qwen-2.5-7B | 0.6 | 1 | 0.0 | | 1.0 | |
| $+\Delta R_{1.5B}$ | 0.6 | 1 | | 1.0 | 0.95 | 16,384 |
| 7B-RLZero | 0.6 | 1 | 1.0 | | | |

**LiveCodeBench** LiveCodeBench [24] is a recently proposed benchmark that aims at a comprehensive and contamination-free evaluation of LLMs for code, which continuously collects new problems over time from contests across three competition platforms, namely LeetCode, AtCoder, and CodeForces. In our experiments, there are 880 test samples in total.

### A.2 Prompt Templates for Inference

Inference prompts used in this work generally following the common practice in the open-source community. Specifically for mathematical reasoning tasks, there are two kinds of prompts as shown in Figure 7. During our experiments, we select experiments with respect to $\Delta R$. For $\Delta R_{1.5B}$, we choose to use *Prompt (b)* following the training and inference setting in [76] since the model scale is too small to follow the complex chat template and output format instruction of "\boxed". For both prompt templates, the "{input}" will be substituted with the corresponding input problem for each data sample.

## B Decoding Configurations

Table 5 presents the detailed decoding configurations for our experiments, specifically, the hyperparameters used.

Table 6: PCR results on various settings.

| Backbone Models | AIME | MATH500 | GPQA | MMLU |
|---|---|---|---|---|
| Qwen-7B | 96.03 | 96.67 | 93.73 | 94.54 |
| Qwen-14B | 95.60 | 96.28 | 93.67 | 93.91 |
| Qwen-32B | 95.22 | 96.58 | 92.93 | 93.32 |
| Llama-8B | 91.15 | 93.16 | 93.23 | 90.44 |

# C   More Results

## C.1   Additional PCR Results

We provide a comprehensive view of PCR results on different datasets, model families, and scales used across this paper in Table 6.

We can see that our conclusion still holds, with over 90 PCRs across all the different model families and different benchmark data points. The results prove the generalization of RAST.

## C.2   Results on Additional Datasets

Apart from the mathematical reasoning and coding tasks explored in Section 3, we also experiment on more general reasoning settings. Specifically, we conduct experiments for RAST on MMLU and GPQA datasets to further demonstrate the generalization on various test settings. Similarly to previous sections, we use primary models from the Qwen-2.5 family across various sales with the corresponding RL-trained variants from SimpleRL-Zoo [76]. The decoding configurations still follow from Appendix B.

Table 7: Experiment results of RAST on general reasoning tasks with the Qwen-2.5-14B and Qwen-2.5-32B models as $\mathcal{M}_{base}$ and $\Delta R$ Simple-RL.

| Models | MMLU | GPQA |
|---|---|---|
| Qwen-2.5-14B | 62.4 | 24.8 |
| $+\Delta R_{7B}$ | 75.1 | 46.7 |
| SimpleRL-14B | 77.8 | 50.3 |
| Qwen-2.5-32B | 61.7 | 38.1 |
| $+\Delta R_{7B}$ | 74.8 | 44.7 |
| $+\Delta R_{14B}$ | 79.6 | 49.0 |
| SimpleRL-32B | 81.4 | 48.3 |

The results are shown in Table 7. We can see that the performance brought by RAST is consistent across both datasets, even if the expert RL model is still trained on mathematical reasoning datasets, which further strengthens the effectiveness and generalization of our method.

## C.3   Performance of Majority@k

We first introduce an additional metric for this additional evaluation.

**Definition of Majority@k**: This metric evaluates accuracy based on the majority prediction among multiple inference runs per problem. Formally,

$$\text{Majority@k} = \frac{1}{N} \sum_{i=1}^{N} \mathbb{I}\left(\text{majority}\{\hat{y}_{i,1}, \hat{y}_{i,2}, \ldots, \hat{y}_{i,k}\} = y_i\right),$$

where the majority function returns the prediction most frequently appearing among the $k$ inference runs for the $i$-th problem. Compared with **Pass@k** that represents diversity, the metric of majority@k reflects the robustness and consistency of the model performance.

We visualize the results in Figure 8, showing the progression of majority@$k$ as $k$ increases across six mathematical reasoning benchmarks. As expected, majority@$k$ improves monotonically with larger $k$, reflecting the benefit of aggregating more diverse sampled trajectories. In most cases, our method (gray bars) bridges a significant portion of the performance gap between the base model $\mathcal{M}_{base}$ (blue line) and the RL-trained model $\mathcal{M}_{RL}$ (red line), particularly on datasets like MATH500 and GSM8K, where our approach nearly saturates the performance ceiling. This suggests strong consistency among sampled outputs and robust transfer of reasoning behaviors.

In contrast, on more challenging datasets such as AIME 2024 and AMC 2023, a wider gap remains between RAST and $\mathcal{M}_{RL}$, indicating that these tasks induce more divergent reasoning paths, where achieving high consensus remains difficult. Nevertheless, even in these harder settings, RAST offers

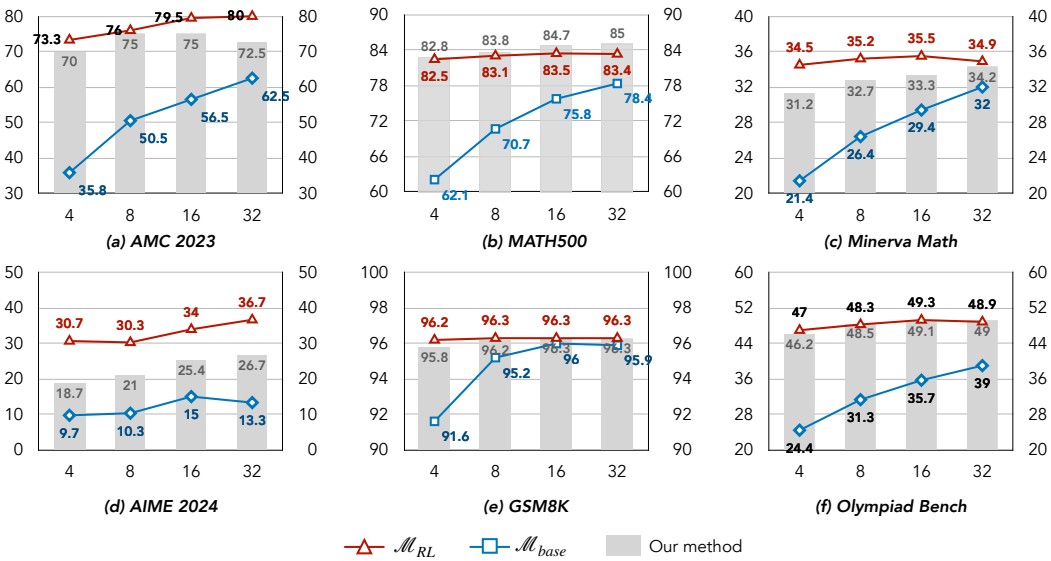

Figure 8: The illustration of majority@k for different values of $k$ on 6 mathematical reasoning datasets, where $\mathcal{M}_{base}$ is Qwen-2.5-32B, RAST uses $\Delta R_{14B}$, and $\mathcal{M}_{Rl}$ is the Rl-trained version of $\mathcal{M}_{base}$.

substantial improvements over the base model, demonstrating that reasoning diversity and consensus can be meaningfully enhanced without full RL training.

## C.4 Response Length

To better understand the behavioral effects of RAST on model generation, we examine the average response length per problem across six mathematical reasoning benchmarks, as shown in Table 8. We report the number of output tokens generated by RAST under each setting.

Firstly, we observe that the RL-trained model generally increases the output length compared to the base models, suggesting that it encourages more verbose or exploratory reasoning paths. This behavior aligns with the goal of RL to promote step-by-step reasoning and self-verification, which naturally results in longer outputs as the model articulates intermediate steps and justifications. We find that RAST inherits this trait: applying logit deltas from RL-trained experts to base models generally leads to increased response lengths, particularly when using deltas from much smaller expert models (e.g., $\Delta R_{1.5B}$). In some cases, the response length under RAST even exceeds that of the corresponding RL-trained model. For example, on AMC and GSM8K, Qwen-2.5-7B augmented with $\Delta R_{1.5B}$ produces significantly longer outputs than both its base and RLZero counterparts, reaching an average of 1362.8 and 354.9 tokens, respectively.

Additionally, we found that the performance of each setting is generally negatively correlated with the length of the generated outputs. Specifically, the length of generated outputs is always the shortest with RAST when we apply $\Delta R_{14B}$ than $\Delta R_{1.5B}$ with 32B base models. More interestingly, we found that the best performance achieved using RAST usually comes with the shortest length, indicating that RAST can achieve both efficiency and effectiveness if configured properly. This also brings up another door for the recent topic that try to compress the reasoning trajectories of RL-tuned models [68, 77, 3, 67, 43].

Overall, these results demonstrate that RAST not only activates reasoning behaviors but also affects the generation style. This controllability opens promising directions for future work in tuning the verbosity and structure of generation by manipulating transfer signals, and supports the broader narrative that RAST serves not just as a reasoning enhancer but also as a tool for decoding-time style modulation.

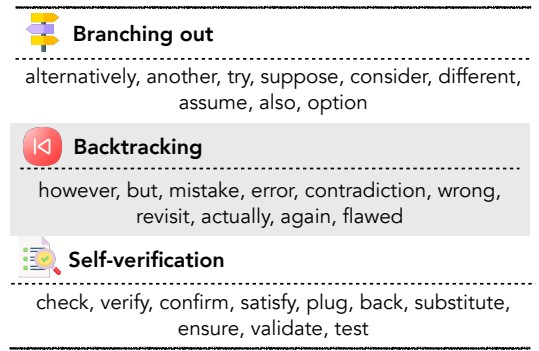

Figure 9: The set of manually curated reasoning tokens that corresponds to three key reasoning behaviours.

Table 8: The response length of output generated per problem by RAST on six mathematical reasoning benchmarks.

| Models | Math500 | AIME24 | AMC | Minerva | Olympiad | GSM8K |
|---|---|---|---|---|---|---|
| Qwen-2.5-32B | 566.9 | 1446.5 | 782.9 | 757.8 | 1103.4 | 275.6 |
| $+\Delta R_{1.5B}$ | 685.1 | 1417.5 | 1024.2 | 799.5 | 978.3 | 333.1 |
| $+\Delta R_{7B}$ | 614.5 | 1033.2 | 895.8 | 635.4 | 870.1 | 307.8 |
| $+\Delta R_{14B}$ | 591.9 | 1164.6 | 863.9 | 605.7 | 886.9 | 297.6 |
| 32B-RLZero | 659.2 | 1183.5 | 940.4 | 655.8 | 992.4 | 307.2 |
| Qwen-2.5-14B | 803.8 | 1138.9 | 820.2 | 921.5 | 1237.0 | 227.5 |
| $+\Delta R_{1.5B}$ | 706.6 | 2092.0 | 1131.9 | 1070.6 | 1438.1 | 337.6 |
| $+\Delta R_{7B}$ | 606.9 | 1092.4 | 919.1 | 636.3 | 889.4 | 309.8 |
| 14B-RLZero | 682.7 | 1477.8 | 1034.9 | 667.1 | 1068.4 | 318.7 |
| Qwen-2.5-7B | 651.3 | 1765.9 | 756.0 | 657.2 | 1033.2 | 263.9 |
| $+\Delta R_{1.5B}$ | 862.7 | 2127.4 | 1362.8 | 944.6 | 1123.5 | 354.9 |
| 7B-RLZero | 682.6 | 1410.5 | 1074.6 | 731.7 | 1033.7 | 330.3 |

## C.5 Case Study on Token-Level Behavior Shift

As a preliminary study mentioned in Section 2.1, we reveal that given the decoding path from RL model $\mathcal{M}_{\text{RL}}$, the base model $\mathcal{M}_{\text{base}}$ actually will largely recover the path, with PRC of $95.22\%$ (for 32B model). In this section, we delve deeper into this study. Firstly, we repeat the study using $\tilde{\mathcal{M}}$ tuned by RAST by feeding the decoding path of $\mathcal{M}_{\text{RL}}$ to $\tilde{\mathcal{M}}$. The PRC in this experiment is $O_{\tilde{\mathcal{M}}} = 96.37\%$, which is even higher. This phenomenon indicates that RAST provides efficient guidance for the base model $\mathcal{M}_{\text{base}}$ in the search space, echoing the conclusion in [74].

Apart from the overall quantitative view, we also present a more intuitive interpretation in Figure 10. We can see that the generated output from $\mathcal{M}_{\text{base}}$ follows only one thinking and contains many erroneous steps without self-verification. However, the output from $\tilde{\mathcal{M}}$ demonstrates a markedly different reasoning behavior. It first *proposes and tests* a candidate solution, then explicitly *verifies* its correctness, and finally reasons about the function behavior to *rule out other possibilities*. To make it more rigorous, we compute the KL divergence [26] online during inference as $KLD(\mathcal{M}_{\text{base}}(t_i, x_{<i}), \tilde{\mathcal{M}}(t_i, x_{<i}))$ where $x_{<t}$ is the current prefix. Notably, we found that the KLD for tokens such as "check" reaches 837.9, which is far larger than normal tokens that usually stay below 1.0. These behavioral differences underscore the effectiveness of RAST in activating reasoning behaviors.

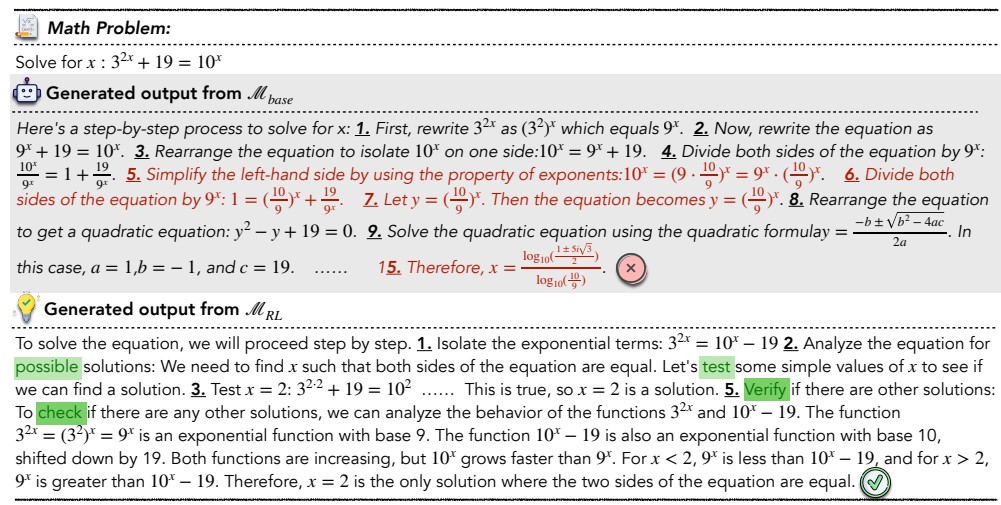

Figure 10: A case study comparing generated outputs for the same math problem sampled from MATH500 from $\mathcal{M}_{\text{base}}$ and $\tilde{\mathcal{M}}$ obtained by RAST. The red denotes erroneous thinking steps from $\mathcal{M}_{\text{base}}$ while the green texts indicate remarkably large KLD, with deeper color denoting larger KLD.

## D    Memory Estimation Details

We estimate the memory considering tensor parallel and CPU offloading (since these are common tricks during training) based on the following dimensions: (i) model memory footprint (FP16), (ii) optimizer states, and (iii) activations & buffers.

**Model Memory Footprint.**    To estimate the memory consumed by model parameters, we consider GRPO training with three model instances: policy, critic, and reference. For each, the parameter size is calculated as (`model size / tensor parallelism factor`) × 2 bytes, assuming FP16 precision. For example, a 14B model with tensor parallelism of 4 would require approximately (`14B / 4`) × `2B` × `3` = `21GB` per GPU. This accounts for only the static model weights without any optimizer states or intermediate activations.

**Optimizer States with CPU Offloading.**    We assume DeepSpeed ZeRO Stage 3 is used to offload optimizer states entirely to CPU memory, which is a common practice in current RL training [79]. Under the Adam optimizer, each parameter typically requires two FP32 states, leading to a total memory footprint of approximately `2× model size × 4 bytes × number of models`. These optimizer states are excluded from GPU memory but are included in CPU memory estimates. For example, in a 14B GRPO setup, this results in approximately 156 GB of CPU RAM usage for the optimizer alone. In our paper, we mainly focus on the GPU memory overheads; therefore, if using CPU offloading, the memory requirement for optimizer states could almost be neglected.

**Activations and Buffer Overhead.**    Activation memory is estimated based on common usage patterns for transformer-based LLMs under long context lengths (e.g., 2048 tokens) and moderate batch sizes. We assume gradient checkpointing is enabled to reduce activation memory, typically resulting in 12–25 GB usage per GPU depending on model size and training configuration. Additionally, we account for 5–8 GB per GPU for auxiliary memory needs such as gradients, residual connections, attention caches, and NCCL communication buffers. These components are summed to estimate total GPU memory usage across devices.

## E    Future Directions

In this section, we briefly discuss the potential future directions following RAST.

**Ensemble Methods.**    One natural extension of RAST involves ensemble strategies [9] across multiple small-scale expert models. Given that each expert model may encode slightly different

reasoning strategies depending on its training trajectory or initialization, aggregating their logit-level deltas could lead to more robust reasoning activation. This ensemble mechanism can be used to reduce variance, enhance generalization across tasks, and potentially adapt to unseen domains with improved resilience. We did a very initial exploration in this direction, by ensembling $\Delta R_{14B}$, $\Delta R_{7B}$ and $\Delta R_{1.5B}$ to the 32B base model of Qwen2.5 on MATH500. We found that the results is actually around 75.6, which does not outperform RAST with a single $\Delta R_{14B}$. We suspect the primary reason is due to the similarity reasoning behaviour encoded in $\Delta R$ across scales in the same Qwen model family. Therefore, the simple ensembling approach will hurt the performance, and also bring much memory overheads for inference. However, we argue that this is still an interesting direction for exploration if we could identify unique and diverse reasoning behaviors of different models before ensembling.

**LoRA from Small Models.** Another intriguing direction is to bridge the output-space corrections from RAST back into the parameter space through low-rank adaptation (LoRA) [21]. Instead of applying logit deltas directly in the decoding phase, we could naturally distil a lightweight LoRA module [80] for each RL-tuned model to imitate the adjustments suggested by smaller RL-tuned models. We may even build reasoning-centric LoRA hub [22] that enables cross-reasoning-behavior generalization and flexible reasoning pattern combination for a more controllable reasoning setting.

**Beyond Reasoning.** While RAST focuses on reasoning activation, the core idea of activating the reasoning behaviors from smaller RL-tuned models via output-space alignment may generalize to other capabilities, such as code generation, instruction following, or factual grounding. Future work may explore how RAST-style activation compares with or complements other alignment techniques, including preference-based fine-tuning or reward modeling.

# F   Limitations

While RAST demonstrates strong empirical performance and introduces a practical paradigm for decoding-time reasoning enhancement, it also comes with several limitations that suggest directions for future research.

**Lack of Theoretical Understanding.** Our method is largely motivated by empirical observations and intuition about how reasoning behaviors are reflected in output distributions. However, the theoretical foundations for why logit-level adjustments from small RL-tuned models can be activated effectively across model scales remain underexplored. A deeper understanding of when and why such activation works — and its potential failure modes — would help establish more rigorous guarantees and improve method design.

**Limited Gains on More Difficult Benchmarks.** Although RAST consistently improves performance across a range of reasoning datasets, the improvements on more challenging tasks, such as AIME, are relatively modest. These datasets often involve abstract reasoning, multi-step derivations, or domain-specific heuristics that may not be sufficiently captured by smaller expert models. This limits the degree of reasoning behavior activation and suggests that RAST may benefit from combining with more targeted adaptation techniques.

**Computational Trade-offs and Expert Quality Dependency.** Despite avoiding full RL fine-tuning on large models, RAST still requires access to a reasonably strong small expert model trained with RL, which can be computationally expensive to obtain. Furthermore, the quality and generalization ability of RAST are inherently tied to the effectiveness of this small expert. If the expert model is poorly aligned or fails to exhibit robust reasoning behaviors, the transferred deltas may provide little benefit.

