# OpenReview forum: "RAST: Reasoning Activation in LLMs via Small-model Transfer"
_NeurIPS.cc/2025/Conference — NeurIPS 2025 poster_

### Official Review · Reviewer_o8y4 · 2025-07-02

**Clarity:** 3
**Significance:** 3
**Originality:** 3
**Rating:** 4
**Confidence:** 4

**Summary:**

The paper proposes RAST, a method to transfer reasoning capability from a small RL-trained model to a larger base LLM without retraining the large model. The authors aim to mitigate the cost of RL training, which only adjusts output distributions on a small subset of reasoning-critical tokens. The authors show that adjustments can be extracted as logit differences between a small model and its RL-enhanced version. At inference time, injecting this difference into the logits of a larger base model can improve reasoning performance.

**Questions:**

Same as weaknesses. The additional questions are:

1. The method proposed in the paper is interesting. An related quesiton is that how the relative size of LLMs influences the performance. For example, whether the difference in a 100M LLM can be used to improve the performance of a 70B model. If not, what is the failure relative size.

2. For robustness of temperature and $\lambda$, it is better to change them jointly. Changing them separately may hide some sensitivity.

**Ethical Concerns:**

["NO or VERY MINOR ethics concerns only"]

**Final Justification:**

The authors' rebuttal largely resolves my concerns. However, since many experimental results are provided in the rebuttal period, the detailed experimental settings are not provided to further justify the quality of these results. Thus, I am toward the borderline of acceptance.

**Limitations:**

yes

**Paper Formatting Concerns:**

N.A.

**Quality:**

2

**Strengths And Weaknesses:**

Strengths:

1. RAST elegantly bypasses costly RL on big models by reusing small-model RL adjustments.

2. The paper verifies an interesting insight: RL only perturbs output probabilities of reasoning-critical tokens.

Weaknesses:

1. The authors mainly focus on the mathematical reasoning problems. However, the **Hypothesis** in the paper does not constrain itself to the math problems. For example, whether the reasoning for general tasks also shares this property is still unknown. Thus, for a logical argument, either experiments in other domains or constraining the argument to math is necessary.

2. Another concern is whether such an argument is influenced by the RL algorithm. For example, whether GRPO/PPO/REINFORCE/RAFT  share the same property is still unknown. The **Hypothesis** should be refined from this perspective.

3. Section 2 is overall too coarse. The presented experimental results cannot support the hypothesis. For example, the different tokens of the base and RL models have not been statistically studied. Figure 10 is hard to understand and interpret, since the normalization procedure is not presented.

---

> ### Author Rebuttal · Authors · 2025-07-31
>
> Thanks for your effort in reviewing our paper and the insightful comments. We appreciate your recognition of our paper’s elegance and interesting insights. Below, we provide detailed responses to each of your comments and hope to address any further considerations you may have:
>
> > **W1: Validation of the Hypothesis on reasoning for general tasks.**
>
> Thanks for the thoughtful review. In the community of RL for reasoning, mathematical reasoning problems are adopted as common practices for evaluation traditions [1,2], so we follow the setting and mainly use mathematical reasoning to test our hypothesis.
>
> To address your concern, we conducted an additional experiment on general reasoning benchmarks, including MMLU and GPQA. Following the experiment setting in Section 2.1, we first calculate the PCR for the Qwen-2.5 series in the following:
>
> |      | MMLU  | GPQA  |
> |------|-------|--------|
> | 7B   | 93.73 | 94.54  |
> | 14B  | 93.67 | 93.91  |
> | 32B  | 92.93 | 93.32  |
>
>
> We can see that the PCR for other general reasoning tasks (MMLU, GPQA) is also pretty high, with over 90%. This indicates that our current hypothesis holds for reasoning generalization.
>
> We then apply RAST to MMLU and GPQA, and the results are shown in the following:
>
> |                 | MMLU  | GPQA  |
> |-----------------|-------|--------|
> | 14B base        | 62.4  | 24.8   |
> | 14B + $\Delta$ 7B  | 75.1  | 46.7   |
> | 14B RL          | 77.8  | 50.3   |
> | 32B base        | 61.7  | 38.1   |
> | 32B + $\Delta$ 7B  | 74.8  | 44.7   |
> | 32B + $\Delta$ 14B | 79.6  | 49.0   |
> | 32B RL          | 81.4  | 48.3   |
>
>
> We can see that the performance brought by RAST is consistent across both datasets, which further strengthens the effectiveness and generalizability of our method.
>
> [1] Wang, Yiping, et al. "Reinforcement learning for reasoning in large language models with one training example." arXiv preprint arXiv:2504.20571 (2025).
>
> [2] Havrilla, Alex, et al. "Teaching large language models to reason with reinforcement learning." arXiv preprint arXiv:2403.04642 (2024).
>
> > **W2: Refine hypothesis w.r.t. different RL algorithms.**
>
> Thanks for raising this point. From a literature perspective, prior work [1] has shown that RL fine-tuning—regardless of the specific algorithm used—tends to result in models that share similar weights with only a small portion of the parameters modified.
>
> From an experimental standpoint, we have also conducted additional analysis using models trained with different RL algorithms from [2] to examine whether similar behavior transfer effects hold in the following table. Note that all the base models here are Qwen-2.5-Math-7B series following [2], and we report the results on MATH500.
>
> |                     | GRPO   | PPO    | (Weighted-)REINFORCE |
> |---------------------|--------|--------|-----------------------|
> | PCR on MATH500      | 97.04  | 96.78  | 97.55                 |
>
>
> PCRs remain pretty high and consistent for RL models trained with a diverse range of algorithms (GRPO/PPO/REINFORCE). Therefore, both conceptually and experimentally, we have sufficient evidence to believe that our hypothesis is generalizable to different RL algorithms.
>
> [1]  Mukherjee, Sagnik, et al. "Reinforcement Learning Finetunes Small Subnetworks in Large Language Models." arXiv preprint arXiv:2505.11711 (2025).
>
> [2] Zhu, Xinyu, et al. "The surprising effectiveness of negative reinforcement in LLM reasoning." arXiv preprint arXiv:2506.01347 (2025).
>
> > **W3: Statistical analysis on different tokens for base and RL models to better support the Hypothesis.**
>
> Thanks for your comment and for reading our Appendix. **Figure 10 in Appendix C.3** is exactly such a statistical analysis for demonstrating tokens related to reasoning behaviors in base, RL, and RAST models. We first manually curate a token list corresponding to three different reasoning behaviors. Then we use Spacy to statistically calculate the token frequencies in model-generated responses. Specifically, we compute the token appearances for each category (branch-out, backtrack, self-verify) in each math problem’s 32 generated trajectories, with the averaged number of appearances per trajectory as normalized frequency. For example, in Figure 10 (a), 1.101 means that tokens belonging to “self-verify” appear 1.101 times on average per problem. As stated in lines 694-700, RAST steers the model behavior from base models by actually changing the corresponding reasoning behaviors.
>
> Tokens are important signals demonstrating reasoning behaviors from models, and are a widely adopted approach for analysis in the community [1,2]. Therefore, we believe our hypothesis and the main conclusions are well justified through the experiments mentioned above.
>
> [1] Gandhi, Kanishk, et al. "Cognitive behaviors that enable self-improving reasoners, or, four habits of highly effective stars." arXiv preprint arXiv:2503.01307 (2025).
>
> [2] Yeo, Edward, et al. "Demystifying long chain-of-thought reasoning in llms." arXiv preprint arXiv:2502.03373 (2025).
>
> > **Q1: How does the relative size between the small and large models affect RAST’s effectiveness?**
>
> Thank you for the interesting question. In our current setup, we use a 1.5B RL-trained model to guide larger base models ranging from 7B up to 32B, and we consistently observe performance improvements across tasks. This suggests that RAST can be effective even when the small model is an order of magnitude smaller than the target model.
>
> In practice, it is commonly observed that RL training becomes more effective at larger model scales [1], while very small models (e.g., ~100M) often struggle to learn meaningful reasoning behaviors via reinforcement learning. In many such cases, even SFT can outperform RL. As a result, directly using a 100M RL-trained model to guide a 70B model is unlikely to yield strong benefits. In this case, those extremely small models can be trained, e.g., via data distillation from a larger RL expert, offering another lightweight and performant alternative.
>
> That being said, our primary motivation is to avoid the high cost of applying RL to large models, not necessarily to minimize the size of the small model. Training a small-sized model, say a 1.5B RL model, is already much cheaper and more accessible, yet still sufficient to capture reasoning behaviors.
>
> Additionally, in a conceptual view, the $\Delta$ captured is used as the key driver of transfer effectiveness. A larger $\Delta$ intuitively tends to indicate more useful reasoning signals, which in turn improves the impact of the guidance on the target model.
>
> [1] Cao, Yuji, et al. "Survey on large language model-enhanced reinforcement learning: Concept, taxonomy, and methods." IEEE Transactions on Neural Networks and Learning Systems (2024).
>
>
> > **Q2: Jointly change for sensitivity analysis.**
>
> Thanks for your constructive comment. We have conducted the experiments jointly, and the results are listed in the following (here we only list a subportion due to character limit in rebuttal; row indicates different $\lambda$ while column indicates different temperatures):
>
>
> |       | 0.8  | 0.9  | 1.0  | 1.1  | 1.2  |
> |-------|------|------|------|------|------|
> | 0.3   | 80.5 | 80.4 | 80.6 | 80.2 | 80.1 |
> | 0.4   | 80.4 | 80.4 | 80.0 | 80.2 | 80.0 |
> | 0.5   | 80.9 | 81.2 | 81.3 | 80.2 | 80.3 |
> | 0.6   | 81.4 | 81.4 | 80.9 | 80.4 | 80.2 |
> | 0.7   | 81.0 | 81.0 | 80.6 | 80.2 | 80.0 |
>
>
> We can see that our conclusion still holds: when the choice is within a certain range for decoding temperature and $\lambda$, RAST demonstrates a robust performance and is insensitive to parameter choices.

---

### Official Review · Reviewer_YtSM · 2025-07-02

**Clarity:** 3
**Significance:** 3
**Originality:** 2
**Rating:** 5
**Confidence:** 3

**Summary:**

The authors propose RAST, a decoding-time method that performs a reasoning knowledge transfer from a smaller to a larger model. The subject of the transfer is what the authors hypothesize to be small, selective adjustments of LM head probabilities. The authors claim that RAST enables large models to match/exceed performance they would otherwise gain by RL on math and coding tasks.
The experiments of the paper are designed to demonstrate that RAST improves decoding diversity and pass@k accuracy, overcoming what the authors claim to be known limitations of RL-based approaches. RAST is positioned as a "new avenue for decoding-time reasoning enhancement".

**Questions:**

See weaknesses.

**Ethical Concerns:**

["NO or VERY MINOR ethics concerns only"]

**Final Justification:**

In the light of the author response I increase my score.

**Limitations:**

No on the limitations and no on the potential negative societal impact, though the latter is not particularly important given the nature of the work.

On the topic of limitations, the paper does include some discussion of the trade-offs in transfer effectiveness, there is no dedicated part of the paper making conceptual outline of the work's/experimentation's limits.

**Paper Formatting Concerns:**

No.

**Quality:**

3

**Strengths And Weaknesses:**

**Strengths**
1. Clear formulation of the cornerstone hypothesis of the paper.
2. Comprehensive, extensive evaluation (Table 2, Table 3, Figure 3), supplemented by clear interpretation of results (Sections 3.2 and 3.3).
3. Contemporary relevance of the work, especially on the front of resource efficiency (1-2x memory and 2x compute reduction with 60-80% average performance recovery; Section 4.4).

**Weaknesses**
1. The authors put the review of the related work at the end of the paper but do not use this to their advantage to make concrete comparisons and links to related work. On this front, I find the current state of Section 5 lacking.
2. I question the relevance of the inclusion of Figure 5 (referenced vaguely from Section 4.2). Perhaps the authors could elaborate on the connection between the Figure 5 and lines 237-245. The current formulation is largely anecdotal and borderline irrelevant, in stark comparison to some other parts of the paper.

---

> ### Author Rebuttal · Authors · 2025-07-31
>
> Thank you for the positive review and constructive comments! We are grateful for your recognition of our paper’s clear formulation, cornerstone hypothesis with comprehensive experiments and clear analyses. Below, we provide detailed responses to each of your comments and hope to address any further considerations you may have:
>
> > **W1: The related work section lacks concrete comparisons and integration with the main discussion.**
>
> Thank you for the feedback. The current related work section is organized based on the unique position of our paper. Our goal is to establish a new paradigm for leveraging small RL-trained models to guide larger models at test time, thus circumventing the high cost of RL training for large models. To the best of our knowledge, there is no existing baseline that directly fits this setup.
>
> Therefore, we choose two subsections of topic-wise — RL training for reasoning models, and technique-wise — decoding-time strategy for behavior steering.
> We will further refine Section 5 to make these contributions more explicit and will contrast our approach with the literature (e.g., training-free alignment) following your advice.
>
> > **W2: The relevance and clarity of Figure 5 are unclear.**
>
> Thanks for the constructive suggestion. We appreciate the opportunity to clarify the purpose of Figure 5 and its connection to Section 4.2 (Lines 237–245).
>
> Figure 5 serves as a concrete, intuitive illustration of the token-level behavior shift we discuss in Section 4.2. While our primary analysis is quantitative (e.g., high PRC scores, KLD spikes on specific tokens), Figure 5 complements this by showing how these shifts manifest in actual model outputs. Specifically, it demonstrates how the RAST-guided model $\tilde{M}$ introduces reasoning-critical steps—such as proposing, verifying, and ruling out possibilities—that are missing from the base model $M_{base}$.
>
> Lines 237–245 describe this exact difference in reasoning patterns: while $M_{base}$​ proceeds deterministically and makes early errors, $\tilde{M}$ engages in more structured reasoning, including self-verification ("check") and hypothesis testing ("possible"). These moments of divergence correspond precisely to tokens with high KLD, as quantified and highlighted in green in Figure 5. This visual alignment between semantic generation and measured divergence provides strong and interpretable support for the behavior shift hypothesis. Therefore, Figure 5 is meant to enhance interpretability by showing exactly what kinds of reasoning behaviors are being introduced by RAST and where. We will revise the text to make this connection more explicit.
>
> > **Limitation: Lack of a dedicated discussion outlining the conceptual and experimental limitations of the work.**
>
> Thanks for your attention to this point. Due to space limitations, we moved the limitation discussion of RAST to the Appendix. Please refer to **Appendix F** for a conceptual outline of limitations both method-wise and experiment-wise.

---

### Official Review · Reviewer_Tjf7 · 2025-07-03

**Clarity:** 4
**Significance:** 4
**Originality:** 3
**Rating:** 5
**Confidence:** 3

**Summary:**

This paper introduces a new framework Reasoning Activation in LLMs via Small-model Transfer (RAST), an efficient alignment technique that transfers reasoning ability from a small reasoning model to larger reasoning model. The method is started from clear motivation and observation that the majority of token probabilities remain even after the base model has been trained with reinforcement learning. It just selectively adjusts the probabilities of certain tokens which influences the model’s deeper reasoning such as branch out, self-verification, and backtracking. Based on the observations, the authors propose RAST to transfer the reasoning capability from smaller reasoning models to larger base models by adding probability differences between rl trained and base model in small model to larger base model. Experiments conducted across various math and coding domains, using models of different sizes and architecture shows that RAST consistently improves the performance without heavily training the large model. Additionally, the paper shows some interesting analysis about the reasoning behavior of LLMs.

**Questions:**

[Q1] Can the method be combined with speculative decoding to accelerate the inference speed? Discussing this could make the method more efficient in terms of both latency and training costs.

[Q2] Is there a way to apply RAST between models that use different tokenizers? For example, from Deepseek-R1 to Llama.

[Q3] Please discuss about the concept of RAST-aware training, which could even further enhance model performance training small models

**Ethical Concerns:**

["NO or VERY MINOR ethics concerns only"]

**Final Justification:**

After reading the authors' rebuttal and further discussions with the authors, I have decided to maintain my score of 5. My main justification is that most of my initial concerns have been addressed, although some issues remain.

**Resolved**

1. Compute budgets of RAST: The authors have demonstrated the computation budget comparison, which can prove the significance of reducing the training computation while increasing the test-time budget.

2. Some discussion about future works: The authors gave a great discussion about interesting future works such as combining with speculative decoding and RAST-aware training.

**Unresolved**

1. Difficulties in using models with different tokenizers: While the authors referenced prior work and suggested potential solutions, these approaches remain suboptimal and are difficult to generalize in practice.

**Limitations:**

yes

**Paper Formatting Concerns:**

Some typos. L150: meaningless hyperlink on text,  L224: (MBase, MRL), L743: (MAHT500)

**Quality:**

3

**Strengths And Weaknesses:**

[S1] The paper is well-written and easy to read with clear visualizations.

[S2] The motivation and the result of observations are intuitive and provide a meaningful way to understand the nature of the reasoning of LLMs.

[S3] The method is well-motivated and demonstrates great performance.

[S4] RAST has the potential to significantly enhance the model efficiency by reducing the training cost.

[W1] Currently, RAST can only be used between models that have the same tokenizer which limits flexibility of the method.

[W2] The method requires 3 models simultaneously during inference time, which necessitates heavy gpu memory and may increase latency.

[W3] Some typos. L150: meaningless hyperlink on text,  L224: (MBase, MRL), L743: (MAHT500)

---

> ### Author Rebuttal · Authors · 2025-07-31
>
> Thank you for the encouraging review and constructive comments! We are happy that you think RAST is intuitive and well-motivated, with great performance and significant potential for efficiency. Below, we provide detailed responses to each of your comments and hope to address any further considerations you may have:
>
> > **W1 & Q2: Applying RAST between models with different tokenizers.**
>
> Thanks for the insightful comment. While we assume the same tokenizers across our experiments, we think that RAST could be applied between models with different tokenizers. The core technique here is tokenizer alignment/mapping across different LLMs, which is another significant research area in the community. So we do not incorporate it in our paper. RAST actually complements those methods, for example, Contextual Dynamic Mapping [1] could be first applied and then we can calculate the distribution for decoding accordingly.
>
> [1] Chen, Yijie, et al. "Enhancing Cross-Tokenizer Knowledge Distillation with Contextual Dynamical Mapping." ACL Findings 2025.
>
> > **W2: RAST requires 3 models simultaneously for inference.**
>
> Thanks for your comment. We acknowledge that our current setup involves 3 models at inference time, which introduces additional memory and compute overhead during inference. However, the inference usage is in exchange for the computationally costly training. We estimate the training and inference FLOPs based on [1] and training hardware/time in [2]. Specifically, for 14B models, they used 16 H100-80G GPUs for 15 hours; and for 32B models, 64 H100-80G GPUs are used for 36 hours. We assume peak FLOPs for H100 as 1979 TFLOPS with 50% FLOPs utilization during training. The results are presented in the following table:
>
> |                      | Training     | FLOPs         | Inference                            | FLOPs/token     |
> |----------------------|--------------|---------------|--------------------------------------|-----------------|
> | 32B RL               | 32B RL       | 8.21*10^21    | 32B RL                               | 6.4*10^10       |
> | 32B base + $\Delta$ 14B | 14B RL       | 8.55*10^20    | 32B base, 14B RL, 14B base           | 1.76*10^11      |
>
>
>
> We can see that the reduced training FLOPs are all worth the increased inference FLOPs. Additionally, the inference stage could be combined with other techniques that can further improve the efficiency, just as you mentioned in Q1, which we will elaborate on later.
>
> [1] Kaplan, Jared, et al. "Scaling laws for neural language models." arXiv preprint arXiv:2001.08361 (2020).
>
> [2] Zeng, Weihao, et al. "Simplerl-zoo: Investigating and taming zero reinforcement learning for open base models in the wild." arXiv preprint arXiv:2503.18892 (2025).
>
> > **W3 & Presentation: Typos.**
>
> Thanks for pointing them out! We have fixed them accordingly.
>
> > **Q1: Potential of combining RAST and speculative decoding to improve inference speed.**
>
> Thanks for the interesting question. We believe there is great potential for RAST to be integrated with speculative decoding for inference acceleration. We provide some initial thoughts here.
>
> One direct way is to use a draft base model to propose tokens (e.g., 7B base model), and then verify them using the full RAST-composed logits. This may reduce the number of full composite forward passes and decoding steps, especially when the draft is accurate.
>
> > **Q3: Discussion on RAST-aware training.**
>
> Thanks for the insightful question. We believe RAST-aware training holds great potential in boosting model performance. We envision several directions in the following:
>
> (i) **RAST-regularized RL training:** One promising approach is to convert the logit deltas and the corresponding branching/backtracking signals into explicit regularization terms during training [1]. These signals can penalize or encourage certain decision patterns, enabling small models to internalize the reasoning behaviors by enforcing trajectory exploration/exploitation during the RL training process.
>
> (ii) **Adapter-based distillation:** RAST outputs/deltas can be used as supervision targets to train task-specific adapters or lightweight finetuning modules (e.g., via LoRA or LoRA Hub-style infrastructure). For the task-specific adapters, speculative decoding [2] would be a good direction to consider. We could also decompose different reasoning behaviors from RAST via LoRA, and achieve controllable and composable reasoning in the style of LoRA-Hub.
>
> (iii) **RAST-enhanced data augmentation:** RAST can be used to generate multiple high-quality and diverse trajectories. For example, we can use test-time scaling to generate new trajectories by sampling on the position of key reasoning tokens. These trajectories—enriched with exploratory behavior and reasoning patterns—serve as additional supervision signals, effectively augmenting the training dataset for training smaller language models to be generalizable and effective.
>
> Overall, we think RAST provides a new paradigm for training efficiency and provides huge potential for follow-up work. We had some initial discussions on Appendix E, and we will incorporate these new insights into the next version of our paper.
>
> [1] Wang, Shenzhi, et al. "Beyond the 80/20 rule: High-entropy minority tokens drive effective reinforcement learning for llm reasoning." arXiv preprint arXiv:2506.01939 (2025).
>
> [2] Fu, Tianyu, et al. "R2R: Efficiently Navigating Divergent Reasoning Paths with Small-Large Model Token Routing." arXiv preprint arXiv:2505.21600 (2025).

---

> > ### Comment · Reviewer_Tjf7 · 2025-08-04
> >
> > Thank you for your rebuttal. While most of my concerns are addressed by your responses, I have some follow-up questions.
> >
> > **[W2-1]** Could the authors provide a comparison of actual inference time, in addition to the reported FLOPs? As the method require additional compute and may hinder parallelization during decoding, this could lead to higher inference latency in practice. Providing this comparison would enable a more constructive discussion and help assess how practical and deployable the proposed method is in real-world scenarios.

---

> > > ### Author Response · Authors · 2025-08-04
> > >
> > > Thanks for reading our rebuttal and for the swift response! We are happy to know that our rebuttal addresses most of your concerns. We also appreciate your suggestion on the comparison of actual inference time, which is indeed a crucial factor in evaluating the method’s practicality. We compute the inference latency in terms of how many tokens are processed per second in the following table with 4 A6000 GPU cards:
> > >
> > > | Model Configuration      | Speed input (tokens/sec) | Speed output (tokens/sec) |
> > > |--------------------------|--------------------------|----------------------------|
> > > | 14B + $\Delta$ 7B               | 65.01                    | 216.12                     |
> > > | 14B RL                   | 75.05                    | 249.54                     |
> > > | 32B + $\Delta$ 7B               | 62.38                    | 202.66                     |
> > > | 32B RL                   | 72.98                    | 243.14                     |
> > >
> > > We observe that RAST incurs around 13–16% latency at inference time compared to directly using larger RL models for decoding. This is expected due to the need to compute multiple model logits at each step. We believe that this latency could be further optimized by efficient decoding methods.
> > >
> > > We will include this latency analysis in the next version of our paper to offer a more comprehensive study on inference efficiency, and thanks again for prompting this important discussion.

---

> > > > ### Comment · Reviewer_Tjf7 · 2025-08-04
> > > >
> > > > Thank you for your detailed discussion. I have no further questions. I believe this paper makes a significant contribution to the NeurIPS community by offering insightful perspectives with interesting analysis. For this reason, I will maintain my score as "accept" (5).

---

### Official Review · Reviewer_zUV7 · 2025-07-05

**Clarity:** 2
**Significance:** 2
**Originality:** 2
**Rating:** 2
**Confidence:** 4

**Summary:**

In this paper, the authors proposed a method called RAST,  trying to enhance the reasoning capabilities of large language models (LLMs) by transferring reasoning behaviors from smaller models trained with reinforcement learning (RL). The authors assume that the changes in output probabilities induced by RL are largely model-size invariant, allowing for efficient reasoning activation in larger models without the need for full-scale RL training, however, this assumption does not have any theoretic support.  RAST involves injecting logit adjustments from a small RL-trained model into larger models, the experimental results show performance improvements on various mathematical reasoning benchmarks.

**Questions:**

I have listed all the weak points of the paper, which are the questions authors need to address.

**Ethical Concerns:**

["NO or VERY MINOR ethics concerns only"]

**Limitations:**

See the weak points listed above.

**Paper Formatting Concerns:**

N.A.

**Quality:**

2

**Strengths And Weaknesses:**

1.The core idea of transferring knowledge from smaller models to larger ones is not entirely new. Similar concepts have been explored in other domains, such as knowledge distillation and model compression. The paper does not provide a clear distinction or significant advancement over these existing methods.
2.The experiments are limited to mathematical reasoning tasks and specific datasets. While the results are promising, the paper does not demonstrate the generalizability of RAST to other types of reasoning tasks or domains. The effectiveness of RAST on more complex or diverse reasoning scenarios remains unexplored.
3.The paper lacks a formal theoretical analysis of why the logit adjustments from smaller RL-trained models can effectively activate reasoning behaviors in larger models.
4.The performance of RAST is highly dependent on the quality of the small RL-trained models. If the small models are not well-trained or fail to capture robust reasoning behaviors, the transferred adjustments may not provide significant benefits. This dependency limits the applicability and reliability of RAST.
5.While RAST avoids full RL fine-tuning on large models, it still requires access to a reasonably strong small expert model trained with RL, which can be computationally expensive to obtain. The paper does not provide a detailed analysis of the trade-offs between the computational cost of training the small expert models and the benefits gained from transferring their reasoning behaviors.

---

> ### Author Rebuttal · Authors · 2025-07-31
>
> Thanks for reviewing our paper. Below, we provide detailed responses to each of your comments and respectfully hope you can consider adjusting the current rating:
>
> > **W1: The idea of knowledge transfer from small to larger models is not new.**
>
> We respectfully believe that this concern may reflect *a misunderstanding of our contribution*.
>
> First, we are not claiming to build a brand-new method for simply transferring knowledge from small to large models. Instead, we first reveal an important insight by introducing and verifying our key hypothesis in line 40. Built upon our insight, we propose a simple and intuitive method to activate reasoning capabilities in large models through a smaller RL-trained model. This way, we can bypass the costly training resource by training large RL models.
>
> Technically, our methodology belongs to the research line of decoding-time steering of model behaviors, and **we have already discussed this research line in detail in Section 5.2**. It differs fundamentally from knowledge distillation or model compression that you mentioned.
>
> > **W2: RAST is only evaluated on math reasoning tasks and specific tasks.**
>
> (i) In the community of RL for reasoning, mathematical reasoning problems are adopted as common practices for evaluation traditions [1,2], so we follow the tradition and mainly use mathematical reasoning to test RAST in a controlled setting.
>
> (ii) Our goal is not to claim a universal solution for all general reasoning problems. We believe that it is unrealistic to expect a single paper to address the full space of general reasoning tasks. Instead, we aim to provide a focused and well-supported demonstration that reasoning activation via test-time RL-guided decoding is effective and efficient—starting with math reasoning as a representative domain.
>
> (iii) We have conducted experiments on domains other than math. In section 3.4, we provide experiments on code reasoning and results in Table 3 demonstrate RAST’s generalizability. During the rebuttal period, we also test on other datasets like MMLU and GPQA. Following the experiment setting in Section 2.1, we first calculate the PCR for the Qwen-2.5 series in the following:
>
> |      | MMLU  | GPQA  |
> |------|-------|--------|
> | 7B   | 93.73 | 94.54  |
> | 14B  | 93.67 | 93.91  |
> | 32B  | 92.93 | 93.32  |
>
>
> We can see that the PCR for other general reasoning tasks (MMLU, GPQA) is also pretty high, with over 90%. This indicates that our current hypothesis holds for reasoning generalization.
>
> We then apply RAST to MMLU and GPQA, and the results are shown in the following:
>
> |                 | MMLU  | GPQA  |
> |-----------------|-------|--------|
> | 14B base        | 62.4  | 24.8   |
> | 14B + $\Delta$ 7B  | 75.1  | 46.7   |
> | 14B RL          | 77.8  | 50.3   |
> | 32B base        | 61.7  | 38.1   |
> | 32B + $\Delta$ 7B  | 74.8  | 44.7   |
> | 32B + $\Delta$ 14B | 79.6  | 49.0   |
> | 32B RL          | 81.4  | 48.3   |
>
> We can see that the performance brought by RAST is consistent across both datasets, which further strengthens the effectiveness and generalizability of our method.
>
> [1] Wang, Yiping, et al. "Reinforcement learning for reasoning in large language models with one training example." arXiv preprint arXiv:2504.20571 (2025).
>
> [2] Havrilla, Alex, et al. "Teaching large language models to reason with reinforcement learning." arXiv preprint arXiv:2403.04642 (2024).
>
> > **W3: RAST lacks a formal theoretical analysis.**
>
> Our work is primarily application-driven, aiming to demonstrate a practical and effective method for transferring reasoning behaviors across model scales. That being said, while a formal theoretical analysis would certainly enrich the understanding of the underlying mechanisms, we believe it is not a prerequisite for demonstrating the utility and novelty of our approach in this setting.
>
> Additionally, we actually already provided a bunch of analyses in the paper, trying to offer insights on “why the logit adjustments from smaller RL-trained models can effectively activate reasoning”. The analyses include PCR (Section 2.1), transferability signal (Section 4.1), and token-level behavior shift (Section 4.2).
>
> > **W4: The performance of RAST is highly dependent on the quality of the small RL-trained models.**
>
> We believe this is not a weakness or limitation, but rather an expected and inherent feature — similar to any machine learning system, the quality of the model’s output definitely is affected by the input to the system.
>
> RAST is designed to be modular and model-agnostic: it can leverage a reasonably good small model that encodes helpful reasoning behaviors, without requiring changes to the large model or costly joint training. This flexibility actually enhances RAST’s applicability, as users can plug in different small experts tailored to their domain or task without retraining the large model.
>
> In our experiments, we show that small RL-trained models across different scales can provide meaningful guidance to much larger base models, resulting in consistent performance improvements across diverse reasoning benchmarks. This highlights the robustness of the transfer/activation mechanism.
>
> > **W5: Lack a detailed analysis of the trade-offs between the computational cost of training the small expert models and the benefits gained.**
>
> **We actually already have such an analysis in Section 4.4**. Results in Table 4 show that with only 50% training resources, RAST can recover more than 80% performance of the trained models. In fact, the efficiency advantage is exactly one of our major claims made in the paper.
>
> **We hope the above responses address your concerns. We are also happy to engage in discussions with you if the reviews could be more constructive and detailed.**

---

> > ### Comment · Reviewer_zUV7 · 2025-08-08
> >
> > Dear Authors,
> >   I have read your rebuttal, however, I still believe that  RAST lacks a formal theoretical analysis, which does not demonstrate the novelty of the idea.
> >
> > Best

---

> ### Comment · Area_Chair_3N74 · 2025-08-05
> **To reviewer zUV7:**
>
> To reviewer zUV7:
>
> - Please read all reviews and author responses, then post a brief reply as early as possible so there is time for meaningful back-and-forth.
> - Join the discussion before clicking “Mandatory Acknowledgement.” Clicking prematurely does not waive this requirement.
> - Reviews without discussion may be flagged Insufficient Review under the Responsible Reviewing rules.
>
> Thank you for helping to uphold the highest standards of the conference.
>
> AC

---

> ### Author Response · Authors · 2025-08-08
>
> Thank you for your response and for taking the time to review our paper.
>
> We would like to respectfully clarify a point regarding your comment:
> > RAST lacks a formal theoretical analysis, which does not demonstrate the novelty of the idea.
>
> We believe that the presence or absence of theoretical analysis is not inherently tied to the novelty of a contribution. Novelty is typically assessed based on the originality and distinctiveness of the proposed idea or method, rather than the specific type of analysis presented.
>
> ---
>
> **1. Novelty of RAST.** RAST introduces a novel inference-time framework that transfers RL-induced reasoning behaviors from small to large models via logit residuals. This approach avoids the need for costly RL training on large models while achieving comparable or better performance. Our method is model-size agnostic, works across architectures, and frames reasoning activation as an inference-time steering problem. To our knowledge, this perspective and approach has not been addressed in prior work. We have also empirically validated our fundamental hypothesis through extensive experiments.
>
> **If there are specific prior works that you believe closely resemble RAST, we would greatly appreciate it if you could point them out**. This would help us better understand your concerns regarding novelty and address them directly.
>
> ---
>
> **2. RAST Is an Empirical Paper.**
> The NeurIPS Reviewer Guidelines state that claims can be supported by **either theoretical analysis or experimental results**. Given that RAST is an application-focused contribution, we have provided comprehensive empirical results and in-depth analyses to support our claims. We believe this is consistent with the expectations for submissions in the Applications track.
>
> ---
>
> In light of the above, we kindly ask if you could reconsider your rating, or alternatively, provide more specific and constructive feedback regarding the aspects of novelty or analysis that you find lacking. Such input would be invaluable for us to further improve the paper. If there are any other concerns that you feel remain unaddressed, please let us know so we can clarify or provide additional information.
>
> Thank you again for your time and consideration.

---

### Official Review · Reviewer_pRmh · 2025-07-19

**Clarity:** 3
**Significance:** 3
**Originality:** 3
**Rating:** 4
**Confidence:** 4

**Summary:**

The authors propose RAST, which replaces RL post-training on large LLMs by RL post-training on smaller LLMs and a simple inference-time steering adjustment. The authors show empirically that RAST often allows large LLMs to perform on reasoning tasks as if they were RL post-trained (or even better), while only incurring the costs of RL post-training on smaller LLMs and a negligible steering adjustment. The authors also show that RAST increases trajectory diversity in the large LLM.

**Questions:**

* It would be useful to have an ablation where the small model is RL trained for the right answer format, and see if RAST still works. Moreover, experiments around the Instruct models mentioned above would be important to understand the method’s validity. Would you be able to kindly provide this?

* It would also be important to have ablations around individual reasoning behaviors to determine which ones are actually important.

* Could the authors explain how RAST outperforms RL on the larger model in Figure 3?

**Ethical Concerns:**

["NO or VERY MINOR ethics concerns only"]

**Final Justification:**

Reading the rebuttals, my technical concerns have been largely mitigated. What holds this paper back is the relative lack of conceptual contribution.

**Limitations:**

yes

**Quality:**

2

**Strengths And Weaknesses:**

# Strengths

* The method is clear, simple, and novel.
* The authors' core insight - reasoning behaviors learnt during RL can be used to elicit better performance from a larger model - is plausible.
* The authors demonstrate empirically that using the single additive logit difference term allows larger models to perform much better on reasoning benchmarks, without additional fine-tuning or weight edits. This works and scales consistently across model sizes and families.

# Weaknesses

* One of the core issues is that the paper never swaps out or ablates the specific behaviors it claims to transfer (e.g. branching, backtracking, self-verification). Experiments that study whether a small model trained to produce the right formatting of answers, or an s1 (https://arxiv.org/abs/2501.19393)-like approach where the small model can force the larger one to answer, or just testing alternative behaviors would be useful. Without such ablations, it is difficult to say whether RAST is truly improving reasoning skills.

* The SimpleRL-Zoo method is used for the small RL teacher and uses MATH500 in its reward data. This also forms the test set, which makes empirical results on the same invalid. It is possible that the improvements could stem from just re-activating memorized solutions rather than transferring actual reasoning skills.

* The experimental setup has additional issues: All comparisons use the base models rather than instruction tuned versions, and show improvements on these base models. Instruction tuning already elicits reasoning capabilities and performance similar to what RAST seems to provide. It would be important to either compare directly against Instruct tuned models or to use RAST on an Instruct model and see if it still improves performance (along with the ablations mentioned above).

* The experiment measuring path-coverage does so on only 50 teacher forced traces from MATH500 on models of the same family. Qwen’s smaller versions contain distillation data from larger versions and have essentially the same training data, so it is unsurprising that these models have similar distributions on benchmark questions. Broad generalisation about “reasoning alignment” seems overstated.

(The reviewer is considering increasing their score if the weaknesses are adequately addressed in the rebuttal)

---

> ### Author Rebuttal · Authors · 2025-07-31
>
> Thanks for the constructive comments. We sincerely appreciate your recognition of RAST's clarity, novelty, and core insights, as well as the empirical validation of its effectiveness and scalability across models without additional tuning. Below, we provide detailed responses to each of your comments and hope to address any further considerations you may have:
>
> > **W1 & Q2: The lack of ablations isolating the specific reasoning behaviors being transferred, and the importance of determining which reasoning behaviors are important.**
>
> Thanks for the thoughtful comment. We agree that demonstrating specific reasoning behaviors being transferred is an important aspect of our claim. Therefore, we conducted such an analysis in **Appendix C.3**. In this analysis, we first manually curate a set of tokens that reflects certain reasoning behaviors as shown in Figure 9. We then statistically test the frequencies of those key tokens as an approximation of appearances for reasoning behaviors. The results are shown in Figure 10. We can see that the reasoning behaviors of RAST are much closer to the corresponding large RL model. We believe that this keyword-based quantitative analysis provides concrete evidence beyond accuracy/performance, indicating that RAST is really steering the base model towards improving real reasoning skills.
>
> We agree with Q2 on the importance of determining important reasoning behaviors. However, this question lies somewhat beyond the scope of our current work, which aims at developing a method for reasoning activation to reduce the costly RL training effort. Determining different reasoning behaviors remains a valuable future direction for follow-up works, as discussed in lines 750-755. As an initial trial, **our analyses in lines 694-700** reveal the correlation between certain reasoning behaviors with specific datasets. For example, AIME 2024 exhibits a pronounced amount of "branching out". This indicates that the AIME dataset needs careful traversing and more sophisticated inductive reasoning capabilities.
>
> > **W2: The use of MATH500 in both training and evaluation raises concerns about whether improvements reflect true reasoning transfer.**
>
> Thanks for the comment. We respectfully think *there is a misunderstanding*. We would like to clarify that **MATH 500 is not incorporated in the reward data** for the small RL model training in SimpleRL. In fact, according to Section 2.1 of the Training dataset in [1], the SimpleRL authors used MATH as the training data, which differs from MATH500. That being said, the training and test datasets do not overlap. So the problem of “re-activating memorized solutions” does not apply in this case.
>
> Despite this, our evaluation is not limited to MATH. Specifically, we evaluate RAST across 5 other diverse mathematical reasoning benchmarks—including GSM8K, AIME 24, AMC23, Minerva, and Olympiad—ensuring that our conclusions are based on a comprehensive and diverse set of tasks, rather than bias or coincidence arising from any single dataset.
>
> > **W3 & Q1: Comparison and integration of RAST with instruction-tuned models.**
>
> Thanks for your constructive comment. We only incorporate base models in our setup because:
> -  It is common practice in the literature of RLVR [2] to evaluate methods on base models (explained as zero-RL in line 67), and
> - The small RL models we used are trained from a base model directly. We intentionally avoid introducing additional instruction-tuned priors to ensure a clean experiment setting for examination of RL-trained models only.
>
> To address your concerns, we additionally conduct experiments on direct inference with instruction-tuned models as a comparison. The results are shown in the following table (we use the instruction-tuned version of Qwen2.5 series):
>
> |       | MATH500 | AIME | AMC  | Minerva | Olympiad | GSM8K |
> |-------|---------|------|------|---------|-----------|--------|
> | 7B    | 70.3    | 8.9  | 49.9 | 23.2    | 33.9      | 89.5   |
> | 14B   | 75.5    | 12.8 | 55.3 | 28.1    | 37.1      | 92.9   |
> | 32B   | 80.3    | 16.6 | 63.7 | 28.1    | 42.5      | 95.4   |
>
>
> Together with Table 1 in the paper, we can see that direct inference with instruction-tuned models does not match the performance of RAST, especially for difficult datasets such as AIME and Minerva. This experiment further strengthens the effectiveness of RAST.
>
> We also conducted statistical analysis on reasoning token behaviors and compared the normalized token frequencies between the instruction-tuned model and RAST following Appendix C.3:
>
> |                     | MATH500 | AIME | AMC  | Minerva | Olympiad | GSM8K |
> |---------------------|---------|------|------|---------|-----------|--------|
> | **32B-instruct**    |         |      |      |         |           |        |
> | Branch out          | 0.39    | 1.10 | 0.70 | 0.25    | 1.03      | 0.06   |
> | Backtrack           | 0.18    | 0.66 | 0.34 | 0.15    | 0.42      | 0.03   |
> | Self-verify         | 1.01    | 1.85 | 1.33 | 0.83    | 1.98      | 0.20   |
> |                     |         |      |      |         |           |        |
> | **32B + $\Delta$ 14B** |         |      |      |         |           |        |
> | Branch out          | 0.46    | 1.41 | 0.75 | 0.30    | 1.35      | 0.07   |
> | Backtrack           | 0.23    | 0.79 | 0.43 | 0.21    | 0.60      | 0.03   |
> | Self-verify         | 1.10    | 2.30 | 1.47 | 0.86    | 2.25      | 0.21   |
>
>
> We found that RAST is better than an instruction-tuned model in terms of reasoning behavior activation.
>
> > **W4: Path Coverage Rate is measured on limited, closely related models.**
>
> Thanks for your insightful comment. We conducted experiments entailing more datasets, including math and general reasoning (AIME, GPQA, MMLU), and an additional model family (Llama 3.1-8B) for PCR calculation. The results are shown in the following table:
>
> |            | MATH500 | AIME   | GPQA   | MMLU   |
> |------------|---------|--------|--------|--------|
> | Qwen-7B    | 96.03   | 96.67  | 93.73  | 94.54  |
> | Qwen-14B   | 95.60   | 96.28  | 93.67  | 93.91  |
> | Qwen-32B   | 95.22   | 96.58  | 92.93  | 93.32  |
> | Llama-8B   | 91.15   | 93.16  | 93.23  | 90.44  |
>
>
> We can see that our conclusion still holds, with over 90 PCRs across all the different model families and different benchmark data points. The results prove the generalization of RAST. We will add these results to Section 2.1 in the newer version of our paper.
>
> > **Q3: Figure 3 interpretation.**
>
> The gray bar in Figure 3 represents the performance by RAST, where the red line indicates the performance of larger RL models. We can see that in all benchmarks except AIME 24, RAST (gray bar) is always above the RL models (red line). On AIME 24, when k grows to 32, RAST chases the performance of larger models.
>
> [1] Zeng, Weihao, et al. "Simplerl-zoo: Investigating and taming zero reinforcement learning for open base models in the wild." arXiv preprint arXiv:2503.18892 (2025).
>
> [2] Guo, Daya, et al. "Deepseek-r1: Incentivizing reasoning capability in llms via reinforcement learning." arXiv preprint arXiv:2501.12948 (2025).

---

### Decision · Program_Chairs · 2025-09-17

**Decision:**

Accept (poster)

**Comment:**

Brief overview of the paper.
RAST is a decoding-time technique that steers a large base model using a small RL-trained model’s logit residuals relative to its base counterpart. The paper motivates this with high path-coverage overlap between base and RL models and reports gains on math and some coding benchmarks, together with a memory-focused efficiency argument.

 Strengths
- Clear, simple mechanism with coherent motivation.
- Consistent empirical gains across model sizes and families on multiple benchmarks.
- Useful analyses (e.g., path coverage; token-level behavior signals) and a practical efficiency narrative focused on avoiding large-model RL training.

 Weaknesses / what is missing
- No direct comparisons against established decoding-time contrastive methods, RAST’s update is mathematically a contrastive logit adjustment, yet there is no direct comparison to DExperts (expert/anti-expert product-of-experts), DoLa (contrasting layers), or Contrastive Decoding on the same reasoning setups, leaving attribution of gains uncertain.
- Potential evaluation confounds (possible train–test proximity on math datasets) and lack of uncertainty quantification; variance across seeds/temperatures/prompts is not reported in the main results.
- Incomplete efficiency picture, memory estimates are provided, but end-to-end latency/throughput under the three-model decoding setup are not benchmarked in-paper.

 Principal reasons for recommendation
While the idea is timely and empirically promising, the absence of head-to-head decoding-time baselines, unresolved evaluation confounds, and missing latency/throughput evidence weaken the strength of the central claims.

 Discussion and rebuttal summary.
Reviewers highlighted missing behavior ablations, possible data overlap, lack of instruction-tuned baselines, related-work integration, tokenizer constraints, and inference overhead. The authors added token-frequency behavior analyses, clarified training data distinctions, provided instruction-tuned comparisons, and later shared preliminary tokens/s numbers indicating modest latency. Some concerns were alleviated, but several outstanding items (direct baselines vs contrastive decoding, formal contamination audits, standardized efficiency measurements) remain only partially addressed.

Decision
Borderline. Slightly above the acceptance threshold, provided that the authors address the missing comparison with contrastive decoding methods: DExperts (expert–anti-expert product-of-experts), DoLa (contrasting layers), or Contrastive Decoding in the final revision.